# Detection of ferrihydrite in Martian red dust records ancient cold and wet conditions on Mars

Adomas Valantinas [1,2] ✉, John F. Mustard [2], Vincent Chevrier [3], Nicolas Mangold [4], Janice L. Bishop [5], Antoine Pommerol [1], Pierre Beck[6], Olivier Poch [6], Daniel M. Applin[7], Edward A. Cloutis [7], Takahiro Hiroi [2], Kevin Robertson[2], Sebastian Pérez-López [2], Rafael Ottersberg[1], Geronimo L. Villanueva[8], Aurélien Stcherbinine [9], Manish R. Patel [10] & Nicolas Thomas [1]

Iron oxide-hydroxide minerals in Martian dust provide crucial insights into Mars' past climate and habitability. Previous studies attributed Mars' red color to anhydrous hematite formed through recent weathering. Here, we show that poorly crystalline ferrihydrite ($Fe_5O_8H \cdot nH_2O$) is the dominant iron oxide-bearing phase in Martian dust, based on combined analyses of orbital, in-situ, and laboratory visible near-infrared spectra. Spectroscopic analyses indicate that a hyperfine mixture of ferrihydrite, basalt and sulfate best matches Martian dust observations. Through laboratory experiments and kinetic calculations, we demonstrate that ferrihydrite remains stable under present-day Martian conditions, preserving its poorly crystalline structure. The persistence of ferrihydrite suggests it formed during a cold, wet period on early Mars under oxidative conditions, followed by a transition to the current hyper-arid environment. This finding challenges previous models of continuous dry oxidation and indicates that ancient Mars experienced aqueous alteration before transitioning to its current desert state.

Identifying the dominant iron oxide phases in Martian dust can provide quantitative constraints on the planet's past and present chemical environments, climate conditions and habitability[1]. On Earth, poorly crystalline ferrihydrite ($Fe_5 O_8 H \cdot nH_2O$) forms through rapid $Fe^{2+}$ oxidation in aqueous environments at circumneutral pH and low temperatures, transforming to crystalline hematite ($\alpha\text{-}Fe_2O_3$) in warm conditions or goethite ($\alpha\text{-}FeOOH$) under sustained water presence[2–4]. These formation and transformation pathways can therefore constrain past environmental conditions including pH, temperature, redox state,

and water availability. The reddish coloration of the dust-covered Martian surface has been investigated since the early telescopic observations that hinted at the presence of impure iron ore known as limonite, which contains the iron hydroxide goethite[5–8]. Meanwhile subsequent ground-based telescopic and laboratory observations attributed the reddish hue to the presence of pigmentary anhydrous hematite (termed "nanophase NpOx") dispersed in the surface regolith and/or coating of rocks[9,10]. Based on the lack of water absorption features at near-infrared (NIR) wavelengths (1–2.5 μm) as determined

[1]Space Research and Planetary Sciences, Physics Institute, University of Bern, Bern, Switzerland. [2]Department of Earth, Environmental and Planetary Sciences, Brown University, Providence, RI, USA. [3]Arkansas Center for Space and Planetary Sciences, University of Arkansas, Fayetteville, AR, USA. [4]Laboratoire de Planétologie et Géosciences, CNRS, Nantes Université, Univ Angers, Le Mans Univ, Nantes, France. [5]SETI Institute, Mountain View, CA, USA. [6]Univ. Grenoble Alpes, CNRS, IPAG, 38000 Grenoble, France. [7]Centre for Terrestrial and Planetary Exploration (C-TAPE), University of Winnipeg, Winnipeg, MB, Canada. [8]NASA Goddard Space Flight Center, Greenbelt, MD, USA. [9]Institut de Recherche en Astrophysique et Planétologie, CNES, Université Toulouse III Paul Sabatier, CNRS, Toulouse, France. [10]School of Physical Sciences, Open University, Milton Keynes, UK. ✉e-mail: adomas_valantinas@brown.edu

by European Space Agency's (ESA) Observatoire pour la Minéralogie, l'Eau, les Glaces et l'Activité (OMEGA) spectrometer[11], it was argued that the anhydrous and dusty regions contain ferric oxides, possibly hematite or maghemite ($\gamma$-Fe$_2$O$_3$). Furthermore, a widely used mineralogical model[11] proposed that these anhydrous ferric oxides in Martian dust formed by continuous oxidation and weathering under water-poor surface conditions during the Amazonian period, which spans from approximately 3 billion years ago to the present.

Early spacecraft observations revealed a distinctive 3-μm hydration feature in the Martian dust spectrum[12,13] well before the weaker NIR spectral features associated with alteration minerals were identified[11]. Later evaluation of the OMEGA data noted that the large 3-μm absorption band is deeper in the observations of bright, dusty regions when compared to dark, less dusty terrains[14,15]. The increased strength of this absorption band in dusty regions was attributed to either higher abundance of water adsorbed on grain surfaces due to the large surface-to-volume ratio of the dust particles[16] or $H_2O$ bound in hydrated minerals in the dust. Using ten years' worth of OMEGA data, it was shown that the 3-μm band can be attributed to tightly bound $H_2O$ and/or hydroxyl groups in the mineral structure of the dust[17]. NASA's Compact Reconnaissance Imaging Spectrometer for Mars (CRISM) also indicated a deep absorption centered at 3 μm in bright, dusty regions[18]. Finally, laboratory reflectance investigations of Martian meteorite ALH 84001 revealed a 3-μm hydration band (due to $H_2O$ stretching vibrations), although no bands were observed at 1.4 μm (combination of OH stretching and H-O-H bending) or at 1.9 μm (combination of $H_2O$ stretching and bending modes)[19]. Basaltic volcanic glasses also typically include a broad 3-μm band due to $H_2O$ without the weaker 1.4 or 1.9 μm features (e.g. ref. 20). In addition, the hydrogen signature in the Martian meteorite "Black Beauty" was attributed to hydroxylated iron oxide minerals[21].

Data collected by the MIMOSII Mössbauer instrument (MB) on the Mars Exploration Rovers (MER) showed the existence of mm-sized hematite spherules (also known as 'blueberries') and goethite in specific rock outcrops as well as the ubiquitous presence of undetermined iron oxide phase ("nanophase NpOx") in the fine dust (e.g. refs. 22,23). While MER MB data can be used to determine the Fe oxidation state ($Fe^{3+}$/$Fe_T$), it is difficult to distinguish the mineralogy of ferric iron present in the Martian dust[24]. This difficulty arises because in the nanocrystal (<10 nm) range, the distinct characteristics of different iron oxides gradually disappear as particle size and crystallinity decrease, resulting in broad and diffuse spectral lines[25,26]. Further, characterization of nanophase components is difficult in mixtures. However, data from the MERs showed that the iron concentration in the fine dust is positively correlated with sulfur and chlorine abundances, whereas dark olivine-rich soils contained lower abundances of these elements, suggesting that iron in the dust is a product of chemical alteration[23,27,28]. The MERs were also equipped with a series of magnet arrays designed to analyze airfall dust. The analysis of the magnetic targets using MB spectra and imaging systems identified two distinct ferric iron endmembers in the dust: one comprising strongly magnetic and dark-colored magnetite, and the other an unidentified bright-colored (oxy)hydroxide exhibiting weak magnetic properties[29,30]. Earlier results from the Mars Pathfinder mission[31], which utilized five magnets of varying strengths, indicated that the magnetic properties of Martian soil are likely due to small amounts of maghemite present in intimate association with silicate particles, suggesting that the dust particles are composites containing both magnetic and non-magnetic components.

NASA's Mars Science Laboratory (MSL) rover provided several key chemistry and mineralogy measurements of Martian dust and soils. The Chemistry and Camera (ChemCam) instrument utilized its laser-induced breakdown spectroscopy (LIBS) capability to analyze the composition of airfall dust. In each of the initial laser shots from a series of 50 shots on dusty rock surfaces and calibration targets that collected dust over the years, ChemCam consistently detected a hydrogen signal that exhibited no diurnal variation, suggesting that hydrogen is chemically bound within the dust particles[32,33]. ChemCam also detected sulfates in the rocks[34] and dust[35] in Gale Crater indicating an ancient evaporitic and acidic environment. Furthermore, samples from the dust-covered sand shadow known as "Rocknest" were measured with the Chemistry and Mineralogy (CheMin) X-ray diffraction instrument. These measurements revealed that up to ~50 wt. % of the material comprising the "Rocknest" scooped soil is X-ray amorphous, with a significant portion (~20 wt. %) of the amorphous component containing iron, though its precise chemical form (whether as oxides, oxyhydroxides, or sulfates) remains unconstrained[36–38]. In addition, the Alpha Particle X-ray Spectrometer (APXS) instrument analyzed air fall dust on the science observation tray. These measurements[39] indicated that the dust is compositionally similar to the bulk basaltic Mars crust[40,41], but is enriched in $SO_3$, Cl and Fe, which is in agreement with MER observations[30]. Both APXS and ChemCam measurements suggested that the amorphous iron oxide component observed in the "Rocknest" soil may be linked to dust[33,39]. The Sample Analysis at Mars (SAM) instrument, which includes a gas chromatograph and a quadrupole mass spectrometer, detected volatile species ($H_2O$, $SO_2$, $CO_2$ & $O_2$) when the 'Rocknest' sample was heated to ~835 °C[42]. This finding suggested that $H_2O$ is bound to the amorphous component of the sample, as the CheMin instrument did not detect any hydrated or hydroxylated minerals in this sample[42]. In conclusion, while MSL's measurements established that iron oxides comprise ~20 wt. % of the X-ray amorphous component in Martian soils, the specific mineralogical nature of these phases has remained elusive.

Here, we present evidence that ferrihydrite (Fe$_5$O$_8$H $\cdot$ nH$_2$O) – a poorly crystalline, hydrated iron oxide mineral – is the dominant iron-bearing phase in Martian dust. Through systematic visible near-infrared (VNIR) spectral analysis, combining orbital and in-situ observations with laboratory studies of submicron ferrihydrite-basalt mixtures, we demonstrate that ferrihydrite provides the best spectral match to the color of Mars. Our spectral analyses show that the addition of Mg-sulfates to ferrihydrite-basalt mixtures reproduces the distinctive 3-micron hydration band observed in Martian dust spectra. Further, to investigate the particle size of Martian dust, we analyzed multi-angular observations from the Colour and Stereo Surface Imaging System (CaSSIS[43]) onboard ESA's Trace Gas Orbiter, which, combined with laboratory spectrogoniometric measurements of ferrihydrite, suggests that dust particles are predominantly submicron (<1 μm) in size. In addition, we show that ferrihydrite maintains its mineralogical structure and does not transform into other iron oxide-hydroxide phases when exposed to simulated present-day Martian conditions (UV irradiation, 6 mbar pressure, $CO_2$ atmosphere). We then discuss its importance and implications for the past climate and habitability on Mars.

## Results

### Spectral match of Martian dust and ferrihydrite-basalt mixtures

The spectrum of Martian dust acquired by CRISM exhibits key spectral features that fit with our laboratory mixture of hyperfine ferrihydrite-basalt powder in a 1:2 weight ratio, measured under ambient conditions (Fig. 1). Both spectra exhibit the following characteristics: (1) a strong absorption at visible wavelengths, (2) a downturn in reflectance near 1 μm, (3) a featureless continuum lacking hydration bands from ~1 to 2.7 μm, and (4) a deep absorption centered near 3 μm (Fig. 1c). Thus, the spectral properties across the 0.4–4 μm visible/near-infrared (VNIR) range of our hyperfine ferrihydrite-basalt mixture provide strong evidence for the presence of ferrihydrite in Martian dust.

Our synthetic ferrihydrite-basalt mixture (Supplementary Fig. 1) shows good agreement in the extended visible range (0.4–1.2 μm) with Martian dust spectra acquired by orbiters and rovers (Fig. 2a). Previous studies[44,45] have demonstrated that the reflectance spectra of

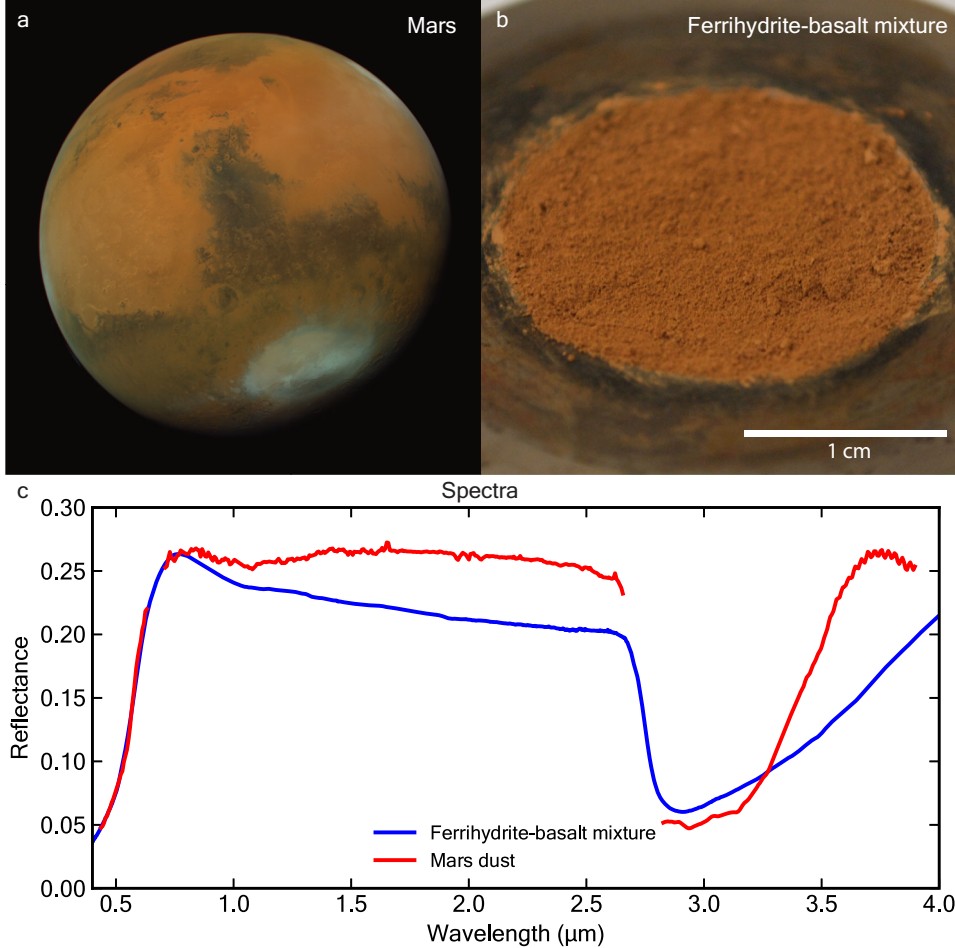

**Fig. 1 | Identification of ferrihydrite in the Martian dust. a** Ochre hue in the light-toned regions of Mars (as observed on 2021-08-14 by the Emirates Exploration Imager; R = 635, G = 546 and B = 437 nm), **b** ferrihydrite-basalt (1:2 ratio) laboratory hyperfine (<1 μm) mixture acquired under ambient conditions in a sample dish, and **c** comparison of an orbital spectrum of Martian dust (from CRISM image FRT00009591) to the spectrum of the ferrihydrite-basalt mixture. The steep increase in reflectance near 0.5 μm is due to the presence of ferric iron and its electron pair transition absorption, which dominates the UV range and extends into the blue wavelengths. The NIR spectral bands at 1.41 and 1.93 μm due to bound $H_2O$ in ferrihydrite are not observed in spectra of these hyperfine mixture samples. The characteristic NIR increase in reflectance (1–2.5 μm) in spectra of pure ferrihydrite (see Fig. 3) is also not observed in our mixture spectra, likely due to nonlinear spectral mixing with the basalt powder. The 3-μm band may be due to chemically bound water in both the Martian dust and the lab sample. Source data are provided as a Source Data file.

ferrihydrite-clay assemblages in the NIR range are consistent with the spectral properties of Martian dust. However, our ferrihydrite-basalt mixtures provide a superior model for the featureless portion of the Martian NIR spectrum, particularly in regions where OH and $H_2O$ bands are typically observed in smectite clay spectra. Our current study provides a comparison of Martian dust spectra from multiple instruments that are all consistent with our ferrihydrite-basalt mixture model. The Martian spectra analyzed for this study include orbital data from CRISM[18] and OMEGA (Supplementary Fig. 2), as well as in-situ measurements from Mars Pathfinder's IMP camera[46], MER Opportunity's Pancam[29], and MSL's ChemCam[47]. Especially telling is the multi-band spectrum of Pancam, which is an observation of the bright and weakly magnetic airfall dust component captured by the Opportunity rover's sweep magnet. The sweep magnet was designed to study weakly magnetic/paramagnetic particles[29], which is particularly relevant as ferrihydrite is paramagnetic under the Martian diurnal temperature range observed at the MER landing sites (-180–300K)[24]. The Pancam spectrum, in agreement with the CRISM observation, suggests a weak absorption feature around 1 μm. This spectral feature is likely due to overlapping absorptions from ferrihydrite and mafic minerals present in the Martian dust, as suggested by the ferrihydrite-basalt mixture spectrum (Fig. 2a, see also Supplementary Fig. 3). Finally, we

note that the dust spectra acquired at five different sites on the planet (Fig. 2a) are spectrally very similar in the visible wavelengths, supporting the hypothesis that Martian dust is well mixed on a global scale[48].

In the extended visible region spectral range (0.4–1.2 μm) iron oxides and (oxy)hydroxides exhibit distinct and diagnostic spectral features[44,49–52] (Fig. 2b, Supplementary Fig. 4). For instance, consider: 1) the deep UV-VIS absorption edge extending from 0.4 to 0.55 μm in the spectrum of hematite, caused by the paired electronic transition $2(^6A_1 \rightarrow {}^4T_1)$, 2) the broad $^6A_1 \rightarrow {}^4T_2$ band at ~0.64 μm in goethite's spectrum and to a lesser extent in spectra of hematite and akaganeite, and 3) the short wavelength ~0.86 μm of the $^6A_1 \rightarrow {}^4T_1$ band in hematite's spectrum. The specific band position and shape[51,53] is a function of the mineral's crystal structure hosting the $Fe^{3+}$. The shape and position of these three bands in hematite-, akaganeite-, schwertmannite- and goethite-basalt mixture spectra are not consistent with the Martian dust spectrum (Fig. 2b). Note that a wide range of weight ratios of hematite-to-basalt cannot reproduce the Martian dust spectrum (Supplementary Fig. 5). In contrast, the spectrum of our synthetic ferrihydrite-basalt mixture replicates the shape and position of the three electronic bands observed in the Martian spectra (Fig. 2a), similar to previous studies of ferrihydrite-smectite assemblages (e.g. ref. 44).

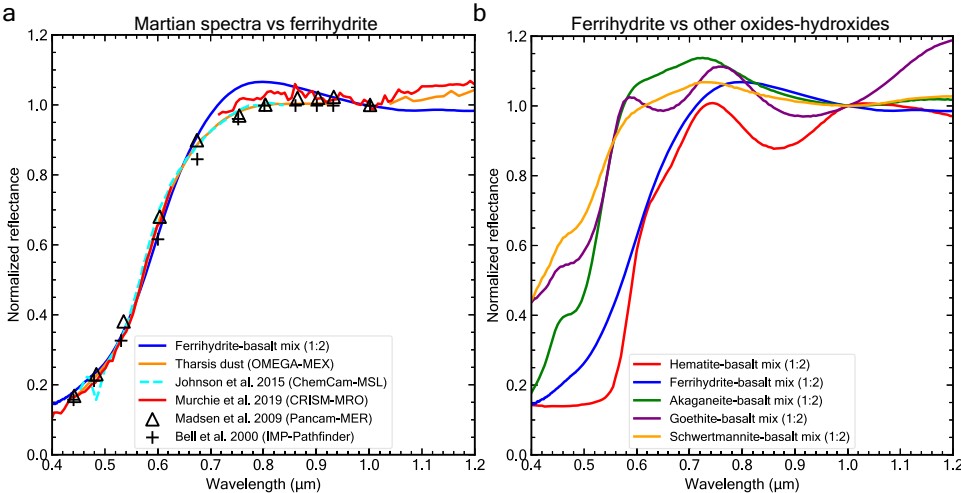

**Fig. 2 | Spectral evidence for ferrihydrite in Martian dust from multiple missions. a** Martian spectra vs lab ferrihydrite-basalt mixture and **b** ferrihydrite-basalt spectrum vs hematite-, schwertmannite-, akaganeite- and goethite-basalt mixture spectra. The ferrihydrite-basalt spectrum provides a better fit to the orbital (CRISM, OMEGA) and in-situ (ChemCam, Pancam, IMP) observations of Martian dust than other iron oxide/hydroxide mixture spectra. Note that other hematite-basalt weight ratios can not reproduce the Martian dust spectrum (Supplementary Fig. 5). The ChemCam-MSL dust spectrum was acquired from the observation of a dusty

rock named "119_Rifle_8" in Gale Crater[47]. The CRISM-MRO spectrum of dust was attained from the Olympus Mons summit observation FRT0000790[18]. The OMEGA-MEX spectrum was collected from the dusty Tharsis region observation orb3741_4. The Pancam-MER multi-band spectrum "B207:Sw, d1" is of airfall dust on the sweep magnet in Meridiani Planum[29]. The IMP-Pathfinder multi-band spectrum was acquired in Ares Vallis and is termed "Surface Dust 21_2"[46]. Source data are provided as a Source Data file.

This suggests that ferrihydrite is a more suitable candidate than other iron oxides and (oxy)hydroxides to explain the ochre hue of Mars (Fig. 1a, b; Supplementary Fig. 6).

Analyses of reflectance spectra of our synthetic mixtures demonstrate that relatively large amounts of ferrihydrite are needed to produce the characteristic red slope (0.4–0.8 μm) of Mars (Fig. 3a, b). 1 wt.% of ferrihydrite in a mixture with basalt does not produce the steep visible slope, nor the prominent visible region absorption at 0.4–0.5 μm. At least 10 wt.% ferrihydrite is needed to exhibit the steep slope, and our analyses indicate that as much as 20-33 wt.% (i.e. 1:4, 1:3 and 1:2 fractions) is needed to provide a good fit with Martian dust spectra. Based on electron microprobe analyses of Fe and O abundances in our synthetic ferrihydrite (Supplementary Fig. 7) and assuming a stoichiometric formula of $Fe_5O_8H \cdot nH_2O$, we estimate that the poorly crystalline iron oxide fraction in our favored 1:2 and 1:3 ferrihydrite-basalt mixtures comprise approximately 21-29 wt.%. This aligns with MSL measurements of total FeO in the amorphous component of the Rocknest sample, which ranges from ~23 to 29 wt.% in Martian soils[37,54]. However, our estimate somewhat exceeds the ~17–20 wt.% $Fe_2O_3$ reported for Martian fines from Viking and Pathfinder missions[55] and the ~18–22 wt.% FeO total in Martian aeolian dust observed by ChemCam[33,35] and APXS[39]. In conclusion, the substantial amount of ferrihydrite in our best spectral analogs supports the hypothesis that part of the significant X-ray amorphous iron component detected in Martian soils by MSL[36,37] may be ferrihydrite.

The featureless portion of the Martian spectrum from ~1.2 to 2.5 μm (Fig. 1c) is consistent with the spectrally neutral properties of the basalt used in the mixtures (Fig. 3a). Variations in slope and continuum in this range may indicate differences in the compaction or induration of the material[56,57]. While our laboratory powders are loose, observations of the Martian surface indicate that dust in some areas may be indurated[58]. In addition, the OH/water absorptions at 1.4 and 1.9 μm are not observed in the ferrihydrite-basalt mixture spectrum. These features are effectively masked because basalt is more absorbent (or less reflective) than ferrihydrite at these wavelengths. The reflectance properties of light, including the degree of multiple scattering and its optical path length, are primarily controlled by the most absorbent material—in this case, basalt. Ferrihydrite, being much more

scattering and much less absorbent, has a reduced influence on these properties.

## Quantitative spectral analysis indicates ferrihydrite

A quantitative assessment based on the root-mean-square error (RMSE; see Methods) calculation suggests that the ferrihydrite-basalt mixture provides a better fit to the observations of Martian dust compared to the hematite-, akaganeite-, and goethite-basalt mixtures (Fig. 4a-d). This is evidenced by the lowest RMSE values of approximately 0.02 for the ferrihydrite-basalt mixture, in contrast to values greater than 0.1 for the other mixtures (Fig. 4e). RMSE estimation provides a single numeric value that summarizes the overall difference between spectra in the normalized reflectance space, with smaller values indicating a closer spectral match between the laboratory and Martian observations.

The small shoulder feature near 600 nm in the Martian dust spectra (Fig. 4a), particularly notable in the ChemCam spectrum may be related to the electron $^6A_1 \rightarrow {}^4T_2$ transition absorption, which is centered around 650 nm (see also Supplementary Fig. 8). This transition is affected by the strength and symmetry of the crystal field, which is dictated by the arrangement of the iron cations' surrounding coordination environment[51]. Our synthetic ferrihydrite-basalt mixture spectrum exhibits less spectral contrast in this range. The contrast of this band is known to vary in natural and synthetic ferrihydrites[59], with some synthetic ferrihydrites lacking this feature entirely. Consistent with these findings, natural ferrihydrite samples collected from a lava cave in Terceira Island, Portugal, and iron-rich springs in Block Island, USA, exhibit observable spectral differences in the 600–700 nm range (Fig. 5). This variability may be due to the sensitivity of ferrihydrite formation in nature to rates of oxidation and hydrolysis, resulting in a range of crystallinities from more disordered 2-line ferrihydrite to the more crystalline 6-line ferrihydrite and many intermediates[60]. X-ray diffraction patterns of our natural ferrihydrites indicate the presence of the 2-line variety (Fig. 5c). However, variations in peak widths (e.g., at ~1.5 Å) suggest some heterogeneity in the crystal structure of the ferrihydrite samples. An alternative explanation to the observed 600 nm feature is that Martian dust may contain small amounts of other (oxy)hydroxides such as goethite, which contain a ~600 nm

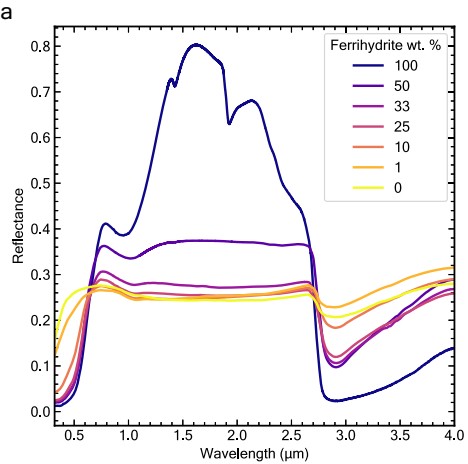
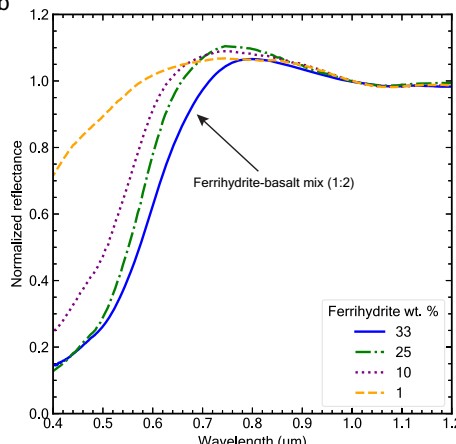

**Fig. 3 | Non-linear reflectance behavior of ferrihydrite-basalt intimate mixtures. a** Ferrihydrite-basalt mixture VNIR spectra with increasing amounts of wt. % ferrihydrite and **b** same spectra normalized at 1 μm and truncated to highlight variations in the extended visible region. The 0 wt. % spectrum represents the pure hyperfine basalt sample. At least 10 wt. % of ferrihydrite is needed to produce the prominent red slope in the visible (0.5–0.7 μm) wavelength range. These spectra suggest that more than 20 wt. % ferrihydrite (i.e. 1:3 weight ratio) may be needed to fit the Martian spectrum of dust. Already at a 1:1 ratio of ferrihydrite to basalt, the NIR (1.3–2.7 μm) increase in reflectance present in spectra of pure ferrihydrite is not observed, nor are the absorptions due to water near ~1.4 and ~1.9 μm. Source data are provided as a Source Data file.

spectral feature. Our mixtures of ferrihydrite, basalt, and goethite suggest that 1 wt. % goethite may explain the observed trends (Supplementary Fig. 9).

### Dehydration experiments suggest ferrihydrite stability

A substantial part of the observed 3-μm band in the Martian dust spectrum (Fig. 1c) may be related to chemically bound water in ferrihydrite. This feature arises from the structurally bound water molecules ($H_2O$) within ferrihydrite's structure[61]. However, ferrihydrite alone cannot fully explain all aspects of the 3-μm band shape observed in the Martian dust spectrum, suggesting the presence of additional hydrated phases. We investigated how these additional hydrated phases might influence the Martian dust 3-μm band shape by creating mixtures with sulfate minerals. The addition of Mg-sulfate to our ferrihydrite-basalt mixture reproduces both the steep slope on the right wing of the 3 μm band and the rollover point at ~3.8 μm observed in the OMEGA dust spectrum (see Fig. 6). This suggests that both ferrihydrite and sulfates may contribute to the hydration signature in Martian dust, consistent with recent MSL ChemCam results that identify sulfates as possible carriers of hydration in Martian soils[35].

Furthermore, under ambient conditions, ferrihydrite also contains some adsorbed water, but most of this adsorbed water is likely removed in the hyper-arid Martian environment. This is supported by a 40-day exposure of ferrihydrite powder to simulated present-day Martian conditions (6 mbar, UV, $CO_2$ atmosphere), which showed a rapid reduction of the 1.9-μm $H_2O$ combination band (see Fig. 7a and Supplementary Figs. 10–12). In contrast, the 1.4 μm band changed little after dehydration. The observation that the 1.4-μm and 3.0-μm bands remain relatively unchanged, while the 1.9-μm band is significantly diminished under conditions of low water vapor relative pressure, is consistent with findings from previous studies[62,63]. Finally, X-ray diffraction (XRD) analyses conducted in our study before and after the experiment indicate that our 2-line ferrihydrite sample did not undergo a phase change due to dehydration (Fig. 7b).

### Orbital and laboratory observations indicate submicron particle size

Our phase curve spectral ratios, derived from CaSSIS observations and laboratory spectrogoniometry, indicate that surface fines including ferrihydrite on Mars may have particle sizes of <1 μm (Fig. 8). Our observations reveal an arch-like curve when reflectance color ratios are plotted against phase angle, a photometric phenomenon previously documented in studies of Martian soils using data from the Viking[64] and MER missions[65]. The roll-over point of both orbital and laboratory <1 μm size fraction curves occurs at ~40° phase angle, suggesting a particle size dependency, as the <11 μm fraction of ferrihydrite exhibits a roll-over point at larger phase angles (Fig. 8b). Alternatively, smooth, semi-transparent, and forward-scattering particles may exhibit an arch-like color ratio curve, while rough particles and their structures may produce a more monotonic reddening[66]. The particle size range derived from these observations agrees with our laboratory submicron mixture samples (Supplementary Fig. 1). Moreover, the particle sizes inferred from our observations and laboratory work are consistent with those derived from atmospheric models[67] and direct microscopic observations of surface fines by the NASA's Phoenix lander[68].

## Discussion

The identification of ferrihydrite as the major oxidized phase in Martian dust raises several important questions: (1) Is ferrihydrite thermodynamically stable under present Martian conditions? (2) What implications does the presence of poorly crystalline ferrihydrite have for the history of water on ancient Mars? (3) What is the nature of surface oxidation on early Mars?

Ferrihydrite is a metastable mineral on Earth that can transform to more thermodynamically stable and crystalline phases over time. There are two main pathways for ferrihydrite to crystallize into other iron oxide phases. The first pathway, known as solid-state transformation, requires high temperatures by dry heating at 200–1000 °C, which causes ferrihydrite to dehydrate and crystallize into hematite[69,70]. However, present-day Martian conditions are characterized by an average temperature of approximately −70 °C and very low water vapor content, resulting from a minimal water vapor partial pressure in the atmosphere[71]. Our 40-day laboratory dehydration experiments, conducted under simulated Martian arid conditions (see Methods), demonstrate that ferrihydrite loses some adsorbed $H_2O$ while maintaining its poorly crystalline structure, as evidenced by X-ray diffraction analysis (Fig. 7a, b). In addition, theoretical constraints[72] suggest that at Martian conditions solid-state transformation of ferrihydrite to goethite is unlikely to be relevant, as the very low average temperature causes kinetic barriers to iron self-diffusion that would require timescales exceeding $10^{16}$ years for even the fastest transformation pathway (see Supplementary Fig. 13). Therefore, both

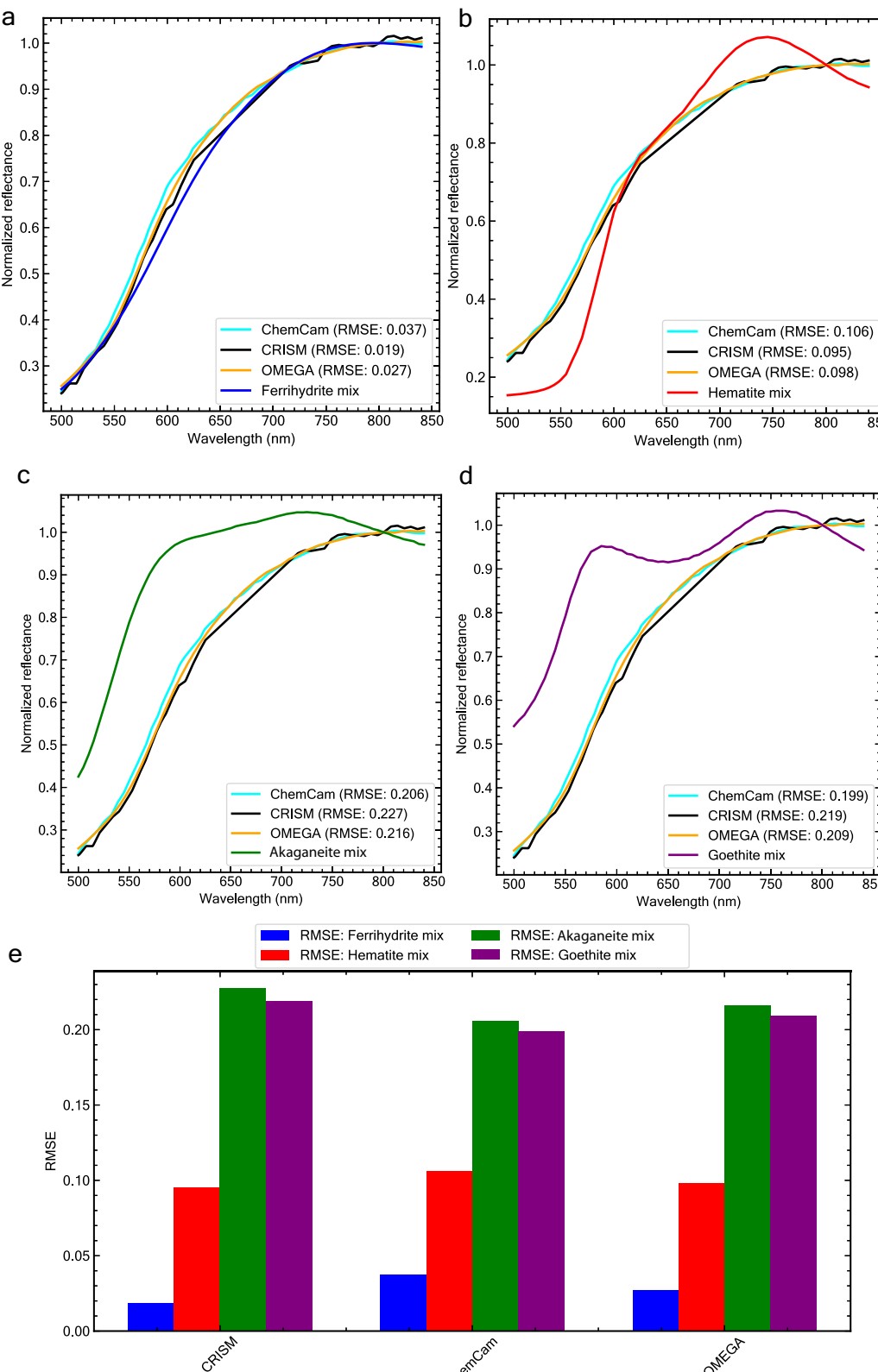

**Fig. 4 | Comparison of observations of Martian dust to lab spectra and RMSE values. a** Comparison of ChemCam, CRISM, OMEGA, and lab ferrihydrite-basalt (1:2) mixture spectra. Mars spectra comparison with the **b** hematite-basalt (1:2) mixture spectrum, **c** akaganeite-basalt (1:2) mixture spectrum, and **d** goethite-basalt (1:2) mixture spectrum. **e** Bar plot of RMSE values derived between each instrument observation and laboratory spectra. RMSE values indicate that ferri-hydrite provides the best fit (low RMSE values) to these Martian spectra. Obser-vations of Martian dust, especially in the ChemCam spectrum, show a small

shoulder at ~600 nm, likely due to the $^6A_1 \rightarrow {}^4T_2$ band at slightly longer wavelengths. Our synthetic ferrihydrite sample lacks this feature but it has been observed in natural ferrihydrite samples[59] (see also our Block Island and Azores natural ferri-hydrites, Fig. 5). An alternative explanation is that there may be minor amounts of other iron (oxy)hydroxide phases present in the dust (Supplementary Fig. 9). The RMSE analysis was conducted over the range 500–840 nm, due to the limitation of the ChemCam data spectral range. All spectra shown were normalized at 800 nm.

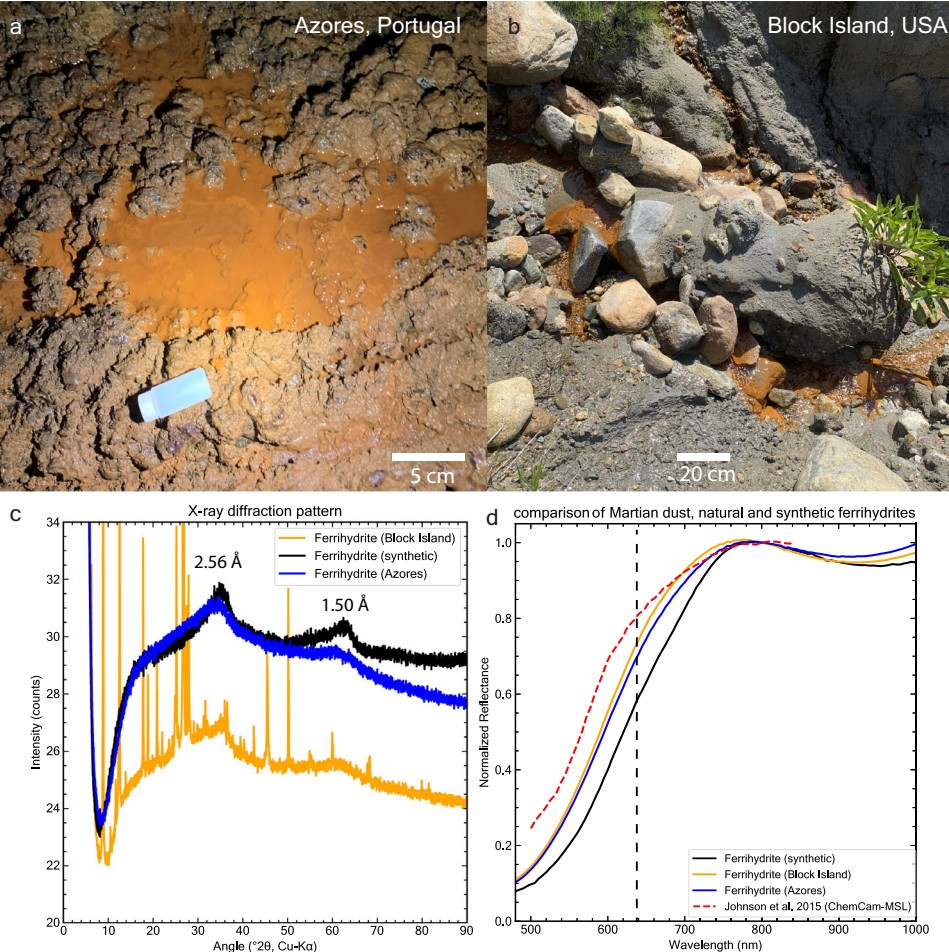

**Fig. 5 | Geologic context and properties of natural and synthetic ferrihydrites.**
**a** Ferrihydrite deposit on a lava cave floor in Gruta Dos Balcões, Terceira island, Azores, Portugal. Iron-rich water was percolating from the basaltic cave ceiling, forming stalactites and depositing on the ground. **b** Ferrihydrite precipitates in a stream in Block Island, Rhode Island, USA. **c** X-ray diffraction patterns of natural and synthetic ferrihydrites used in this study. The peaks at 2.56 Å and 1.5 Å are characteristic of 2-line ferrihydrite, demonstrating its poorly crystalline structure[4].

Ferrihydrite from Block Island is impure and contains quartz ($SiO_2$), some illite, chlorite and traces of other minerals. **d** VNIR spectra showing the variability of natural and synthetic ferrihydrites in the $^6A_1 \rightarrow {}^4T_2$ band position (dashed black line), compared to ChemCam LIBS spectrum of dust[47] and the resulting spectral shoulder at the 600–650 nm wavelength range. Spectra normalized at 800 nm. Source data are provided as a Source Data file.

theoretical and empirical evidence suggests that the cold and dry conditions on present-day Mars do not favor the solid-state transformation of ferrihydrite.

The other pathway is reconstructive transformation, which involves dissolution and precipitation in aqueous solutions, usually resulting in the crystallization of goethite and then hematite. This process is heavily dependent on pH and water temperature; low pH values (high acidity) and low water temperatures tend to slow down this conversion[2,73,74]. Using the Arrhenius equation and the first-order reaction rate constants from Das et al.[74], we estimate that a sample consisting entirely of ferrihydrite would take approximately 1000 years to fully crystallize into hematite under conditions of pH 2 and a water temperature of 1 °C (see Supplementary Information). However, organic compounds[75], silicates[54,60,76], and phosphates[77] have been shown to inhibit or at least slow down the transformation of ferrihydrite to other phases. In addition, under a cold and wet Mars scenario (e.g. refs. 78,79), where liquid water temperatures may reach extreme lows of −30 °C in acidic and saline conditions, the reconstructive transformation could be further delayed. Both acidic and low-temperature conditions required for ferrihydrite's thermodynamic stability may have been present on early Mars, as suggested by observations of the minerals jarosite with in-situ rover experiments (e.g. refs. 80–82) and alunite from orbiters[83]. Consequently, the

reconstructive transformation of ferrihydrite might have been significantly slowed or even prevented in the aqueous environments of ancient Mars.

In nature ferrihydrite is often found admixed with other minerals containing Si and/or $SiO_2$ (e.g. ref. 84) (also see our samples from Block Island, Fig. 5c). Our laboratory experiments, which involved mixtures of silicate (basalt) and ferrihydrite yielded spectral signatures consistent with those observed on the Martian surface (Fig. 1). This correlation suggests that ferrihydrite on Mars is likely not present in its pure form but rather exists in association with volcanic materials, particularly basalt. The presence of a basaltic component in the dust may imply two different genetic relationships: 1) it may indicate that ferrihydrite formed in an aqueous environment associated with basalt, or 2) that the basalt is derived from unaltered regions through physical weathering. The strong winds and prevalent aeolian processes on Mars would support global abrasion and mixing of loose basalt particles with fine-grained ferrihydrite. The fine grain size of ferrihydrite provides additional constraints on its formation conditions. When ferrihydrite precipitates from solution, it initially forms very small crystallites of 2–3 nm[85], likely due to rapid precipitation or hydrolysis, but after drying forms larger aggregates[3,53]. The ~500 nm size of ferrihydrite in our study and its match to Martian spectral data suggests aggregation of these nano-crystallites may have occurred under

similar rapid precipitation conditions, followed by drying. This is consistent with brief episodes of aqueous alteration rather than prolonged weathering, which would typically produce more crystalline phases. Furthermore, it is important to note that on Earth ferrihydrite is often found in volcanic settings such as lava caves (see our Azores sample; Fig. 5a) and tephra[86–88]. Ferrihydrite readily forms as one of the first poorly crystalline phases from alteration of Andosols, alongside allophane and imogolite (e.g. ref. 89). The relative abundance of these minerals depends on the chemistry of the volcanic material. In more Fe-rich volcanic material, ferrihydrite-like species tend to dominate, while allophane and imogolite are less prevalent[90]. Furthermore, the early alteration of basaltic glass primarily produces iron hydroxides with minor amounts of imogolite and kaolinite, rather than smectites which typically form during prolonged Andosol development under warmer and wetter conditions[86].

The presence of ferrihydrite on Mars may provide crucial insights into the episodes of aqueous and oxidative weathering during the planet's early history, potentially occurring at near-freezing surface temperatures. This formation mechanism aligns with climate and geochemical models characterized by fluctuations between above-freezing conditions, facilitating the melting of surface ice deposits, and intermittent colder, arid periods[91,92] (see also Fig. 9). Such environmental conditions that followed the significant period of formation of clay minerals[93] could have existed during the late Hesperian period, approximately 3 billion years ago. In the Hesperian epoch there was a phase of intense volcanic activity (e.g. ref. 94) that could have interacted with liquid water or ice and produced conditions favorable for formation of ferrihydrite. However, the prevalence of amorphous material including allophane[91,95] and ferrihydrite suggests only brief periods of aqueous activity, as ferrihydrite formation necessitates rapid kinetics. This implies that ferrihydrite formation on Mars was likely a geologically swift process[49], contrasting with slower, continuous processes such as gas-solid or photochemical weathering[11,96]. The presence of sulfates and ferrihydrite in the dust, indicated by our 3-μm band analysis (Fig. 6), aligns with the transition toward more acidic and arid conditions late in the Hesperian period. Finally, the ubiquitous ochre hue observed on the contemporary Martian surface, likely due to the widespread presence of ferrihydrite, suggests an extended period of aeolian erosion and global redistribution through wind activity and dust storms. This scenario implies the long-term stability of ferrihydrite on the Martian surface.

The question about the nature of surface oxidation on Mars is important in view of widespread abundance of ferrihydrite in the surface fines and dust. In Fig. 10 we show the redox stability field of ferrihydrite, showing that oxidizing conditions are required for its formation. In aqueous solutions, ferrihydrite forms through two pathways either involving the hydrolysis of $Fe^{3+}$ or the oxidation of $Fe^{2+}$. In laboratory settings, the hydrolysis of $Fe^{3+}$ salts (such as ferric nitrate) leads to the formation of ferrihydrite[53]. As the rate of hydrolysis increases, the crystallinity of the resulting ferrihydrite decreases, which is evidenced by a reduction in the number of X-ray diffraction peaks from 6-8 to 2[97]. However, in the Martian context, oxidation of

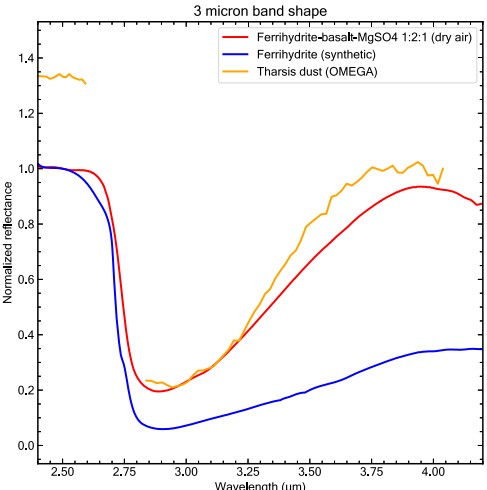

**Fig. 6 | 3-μm band shape comparison for sulfate-bearing mixture, pure ferrihydrite and Martian dust.** The spectral comparison reveals that the 3-μm band's long-wavelength wing and particularly its inflection point near 3.8 μm in our ferrihydrite-basalt-sulfate mixture matches the Martian dust spectrum, suggesting the possible presence of sulfates in the Martian dust. Reflectance spectra of both the ferrihydrite-basalt-sulfate mixture and pure ferrihydrite were measured under ambient conditions with dry air purging immediately before each acquisition. Normalized at 2.5 μm. Source data are provided as a Source Data file.

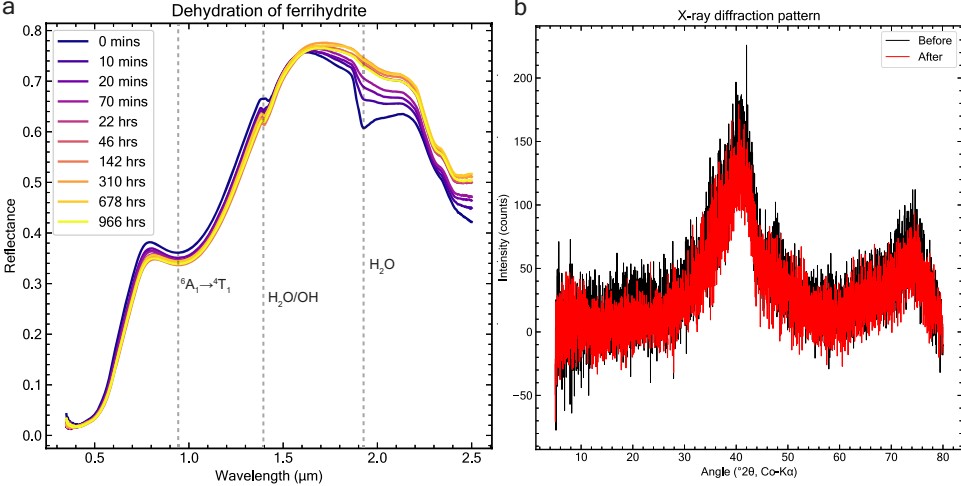

**Fig. 7 | 40-day dehydration experiment of pure ferrihydrite. a** Spectra of pure ferrihydrite under dehydrating conditions for almost 1000 h and **b** background-removed 2 L ferrihydrite X-ray diffraction patterns of before (black) and after (red) dehydration. In **a** exposure to simulated present-day Mars-like conditions results in significant dehydration and water loss in ferrihydrite, as indicated by the removal of the 1.9-μm band. However, the pure ferrihydrite sample did not change phase nor crystallize as suggested by the XRD analysis in **b**. The small spectral shift in **a** from 1.41 to 1.40 μm occurs as adsorbed water (which absorbs light at slightly longer wavelengths) is removed from the system, leaving only structural OH/$H_2O$ groups contributing to the absorption band[123,124]. Source data are provided as a Source Data file.

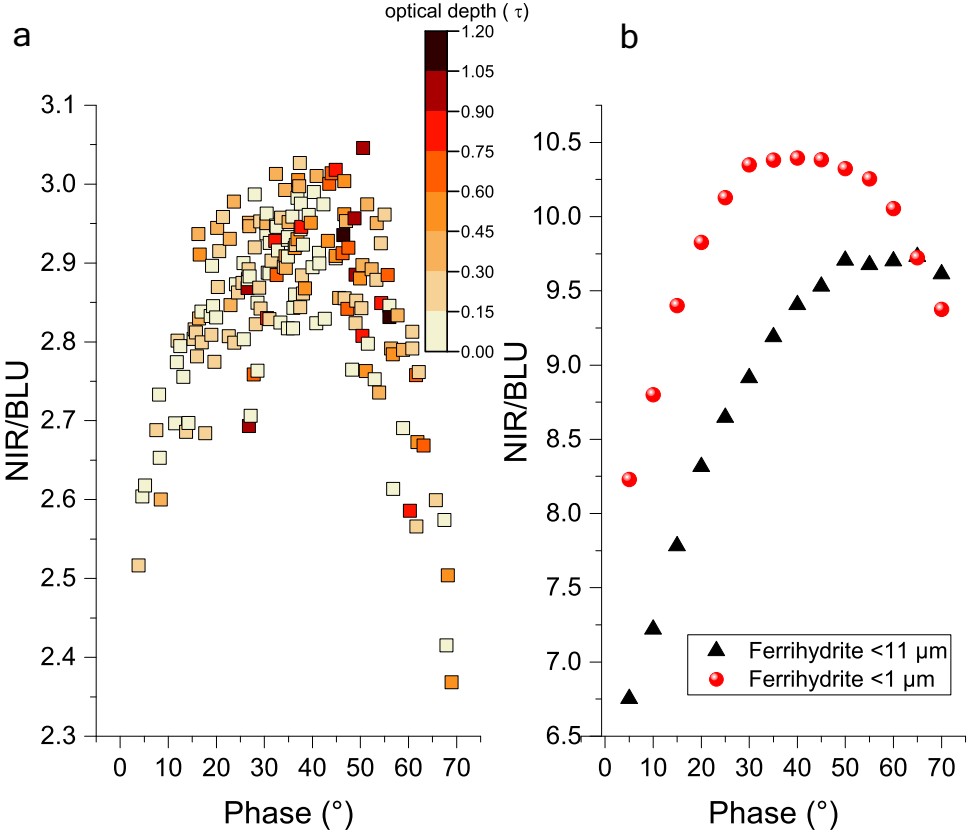

**Fig. 8 | Orbital observations of Martian surface dust and comparison to laboratory spectrogoniometer measurements. a** Phase curve reflectance ratio in two CaSSIS filters (NIR-940 nm, BLU-479 nm) of observations of dust in Arabia Terra and **b** same ratios for two sieved synthetic ferrihydrite size fractions. The observed arch in both plots is known as the phase reddening and bluing effect. The color ratios of both the <1 μm size fraction and Martian observations exhibit maxima at approximately 40° phase angle, followed by a distinct decrease at larger angles. In contrast, the <11 μm size fraction lacks a clear inflection point, suggesting

particle size-dependent variations in light scattering properties. These observations imply that particle size and/or physical particle properties of Martian dust are similar to <1 μm ferrihydrite particles. Optical depth overall is low in (**a**) implying that atmospheric effects are not the cause of the trends observed here. Similar reflectance ratio curves are observed for other dusty regions of Mars[125], which suggests homogenous mixing of dust on a global scale. Source data are provided as a Source Data file.

ferrous $Fe^{2+}$ is more likely due to the large abundance of primary Fe-bearing silicate minerals such as olivine or pyroxene (e.g. refs. 98,99). The key question is whether the $Fe^{2+}$ was oxidized by components in the atmosphere or by processes occurring in surface waters on Mars. Experiments and geochemical calculations have shown that the oxidation of $Fe^{2+}$ and precipitation of crystalline (oxy)hydroxides could occur in shallow waters through two main pathways: either using oxygen atoms produced by the breakdown of atmospheric gases (atmospheric photolysis[100]), or through chemical reactions driven by ultraviolet light interacting with water molecules (water UV photooxidation[100–102]). The latter process, producing reactive species such as hydroxyl radicals (OH), may be more efficient due to lower energy requirements than atomic oxygen formation and could serve as potent surface oxidant. However, chemistry experiments also demonstrated that even in the absence of UV radiation or free oxygen, iron oxidation can proceed in $CO_2$ and $H_2O$ atmospheres[103,104]. These experiments showed the formation of ferric (oxy)hydroxides such as goethite and lepidocrocite, as well as ferrous minerals including siderite ($FeCO_3$) but no ferrihydrite. A two-stage model was proposed[103] where early weathering produced siderite, which then oxidized to $Fe^{3+}$ phases as hydrogen escaped to space, suggesting a self-sustaining oxidation process that could operate without atmospheric oxygen. In contrast, the oxidation of surface rocks on early Mars by atmospheric oxygen has been suggested by recent studies of Gale Crater, which identified high manganese oxide enrichment in lacustrine sediments[105,106]. However, alternative oxidants of the manganese

deposits have been proposed such as perchlorate, chlorate, bromate, and nitrate[107]. Chlorate also has been identified as a possible $Fe^{2+}$ oxidant but it led to the formation of more crystalline iron phases in lab experiments instead of ferrihydrite[108]. On Earth, the majority of atmospheric oxygen is derived from biological activity, making it crucial to study this question on Mars to understand the potential for past life and the planet's habitability. Finally, while previous studies provided valuable insights into potential Mars oxidation processes, the widespread presence of ferrihydrite in the Martian surface materials remains a matter of debate, underscoring the need for further research to clarify the redox conditions leading to ferrihydrite's formation on Mars.

In conclusion, our findings indicate that ferrihydrite, a hydrated iron oxide-bearing mineral, is the major iron oxide component of Martian dust, contrary to previous assumptions of anhydrous minerals dominating the surface. This hydrated phase likely formed during cold water activity in the last stages of early Mars, under oxidative conditions. The widespread presence of ferrihydrite suggests that Mars subsequently transitioned to a hyper-arid, erosional environment that has persisted to the present day, preserving this mineral phase across the planet's surface. The future Mars Sample Return (MSR) mission may deliver regolith samples to Earth that potentially contain ferrihydrite-bearing materials. Detailed analyses of these samples, including Fe, H, and O stable isotope measurements, will provide crucial information about redox conditions, habitability, and water activity on Mars.

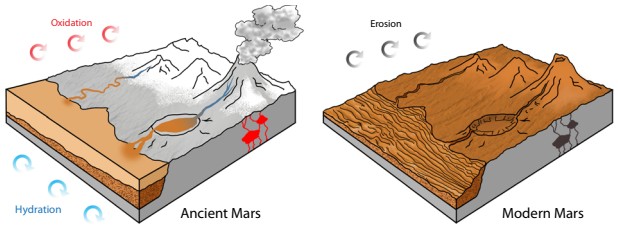

**Fig. 9 | Comparison of major surface processes on ancient and modern Mars.** The left panel depicts ancient Mars during a period of active chemical weathering through hydration and oxidation of basaltic crust to produce ferrihydrite-rich waters. Meltwater runoff triggered by volcanic activity transported insoluble ferric iron into crater lakes and basins to form sedimentary deposits. The right panel shows modern Mars where continuous erosional processes rework sedimentary layers and distribute fine-grained material across the planet to create its characteristic ochre appearance. Schematic not to scale.

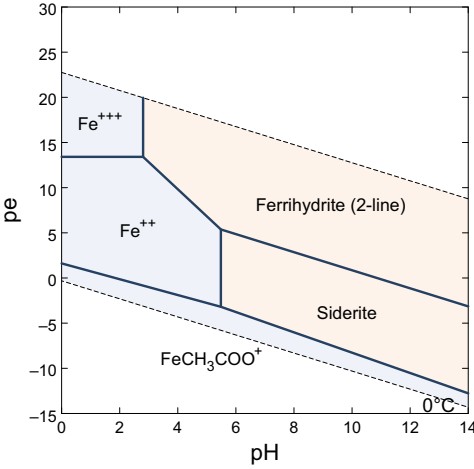

**Fig. 10 | pe-pH predominance diagram (Pourbaix diagram) showing iron species distribution as a function of redox potential (pE) and pH.** 2-line ferrihydrite is the least crystalline type of ferrihydrite, which includes two broad peaks in X-ray diffractograms. While the diagram shows ferrihydrite is thermodynamically stable across a wide pH range, kinetic factors favor its formation at circumneutral pH (6–8) in both natural environments and laboratory synthesis. Constructed using the Geochemist's Workbench software package and the Thermo.com.v8.r6+ database available with the software. The database was updated to include the thermodynamic parameters (solubility constants as a function of temperature) for ferrihydrite (2-line) and ferrihydrite (6-line)[126]. The activity of iron in the liquid phase was set at $10^{-3}$, the $CO_2$ partial pressure at 1 bar and the temperature at 0 °C.

## Methods

### Laboratory sample preparation

Nine different iron oxide phases were used in this study: magnetite, maghemite, goethite, hematite, ferrihydrite, lepidocrocite, akageneite, feroxyhyte, and schwertmannite (Supplementary Fig. 6). Maghemite, ferrihydrite, akageneite, feroxyhyte, and schwertmannite were synthesized at the University of Arkansas following the procedures of Schwertmann and Cornell[53], and were kept under ambient conditions for 14 years before being investigated in this work. The samples were not altered in any way and their XRD patterns can be seen in Supplementary Fig. 14. For ferrihydrite specifically, the synthesis procedure is as follows: 40 g of $Fe(NO_3)_3 \cdot 9H_2O$ was dissolved in 500 mL distilled water, then we added 330 mL of 1 M KOH solution to bring the pH around 7-8, while stirring the solution. The last 20 mL of KOH solution were added drop by drop. The samples were then centrifuged, then rinsed several times with DI water and dried at 50 °C.

For akaganeite synthesis, 54.06 g of $FeCl_3 \cdot 6H_2O$ was dissolved in 2 L distilled water (0.1 M solution) and held in a closed flask at 40 °C for 8 days, yielding approximately 5 g of product that was subsequently washed and dried. Schwertmannite was prepared by dissolving 10.8 g $FeCl_3 \cdot 6H_2O$ and 3 g of $Na_2SO_4$ in 2 L of distilled water preheated to 60 °C, followed by 12 minutes of additional heating at the same temperature. After cooling to room temperature, the suspension was filtered, washed and dried. The presence of natrojarosite in the XRD spectra was noted as a natural co-precipitate during this reaction, possibly due to the omission of dialysis.

Feroxyhyte synthesis began with preparing 300 mL of 0.1 M FeII solution using $FeCl_2 \cdot 4H_2O$ in distilled water, which was filtered to remove oxidized products. The solution's pH was raised from 3 to 8 using dropwise addition of 5 M NaOH under stirring, resulting in green rust precipitation. Addition of 40 mL of 30% $H_2O_2$ solution caused instantaneous oxidation, forming a reddish brown precipitate. The pH was readjusted to 8, and the precipitate was filtered, washed and dried. Maghemite was synthesized by preparing a 0.064 M mixture of $FeCl_2$-$FeCl_3$ with a FeII/FeIII molar ratio of 9, which was then slowly oxidized with an airflow of approximately 10 mL/min in a closed flask at 20 °C.

The magnetite powder is from the University of Bern geological collection and goethite is a natural sample from the mine of En Bournegade, Tarn, France. The lepidocrocite sample is from Siegen, Germany. The hematite powder (<1 μm particle size as shown in Supplementary Fig. 1), was purchased from Sigma-Aldrich Co. The basalt sample used in this study is from the University of Bern geological collection and comes from an unknown locality in Virginia, USA. However, we include mineralogical analysis of this sample as shown in Supplementary Fig. 15. The natural ferrihydrite samples shown in Fig. 5 are from a lava cave in Gruta Dos Balcões, Terceira island, Azores, Portugal and were collected under permit CCIR-RAA/2023/43 issued by Direção Regional da Ciência e Tecnologia, Região Autónoma dos Açores. The ferrihydrite from the iron-rich spring was collected in Block Island, Rhode Island, USA. Both natural ferrihydrite samples were collected as aqueous suspensions and subsequently air-dried under ambient conditions. The Azores 2-line ferrihydrite remained poorly crystalline, showing no phase transformation even after 6 months in solution. pH measurements of the source water yielded circumneutral values (~7) for both Block Island and Azores samples.

The sample preparation procedures differed between samples. For seven of the nine iron oxide phases (magnetite, maghemite, goethite, lepidocrocite, akageneite, feroxyhyte, and schwertmannite), preparation consisted of a single size-separation step. To remove large grains and homogenize our samples using a dry sieving procedure, we used a sonic separator VariSifter™ (Advantech Manufacturing, Inc.) at the University of Bern. Because cohesiveness properties varied across all samples it was not always possible to sieve to the same particle size, so the mesh size had to be varied and experimentally determined for each case. We used mesh sizes of 40- 30-, 20- and 10-μm. Particle sizes were: goethite (<11 μm), lepidocrocite (<40 μm), akaganeite (<30 μm), maghemite (<20 μm), schwertmannite (<30 μm), feroxyhyte (<40 μm), and magnetite (<10 μm).

The remaining two samples, ferrihydrite and basalt, underwent additional grinding to simulate Martian dust particle size. This process was carried out at the University of Grenoble using a Planetary Grinder Retsch© PM100, following similar procedures to the ones defined by ref. 109. The grinder uses a 25 mL zirconium oxide ($ZrO_2$) bowl filled with equal volume fractions of $ZrO_2$ grinding balls, sample powder, and free space. The initial sample powders made of grains smaller than 50 μm were first dry-ground for 20 min with 2-mm grinding balls. Then, the sample powder was recovered and wet-ground for 150 min in ethanol with 500-μm grinding balls. After grinding, the balls were separated from the sub-micrometer-sized sample powder by sieving using additional ethanol. The suspension of sample powder in ethanol was centrifugated and the ethanol evaporated in ambient room

temperature under a hood to obtain the final dry powder. The resulting sub-micron particle size (<1 μm) was confirmed by scanning electron microscope (SEM) images (Supplementary Fig. 1). Given the extensive time required for the specialized grinding procedure and the primary focus on comparing ferrihydrite and hematite as potential major components of Martian dust, we prioritized these two phases for submicron size preparation. The other iron (oxy)hydroxides with particle sizes ranging from <40 μm to <10 μm were included to demonstrate their distinct spectral properties and rule out their dominance as the primary iron-bearing phase in Martian dust, which could be accomplished effectively with their standard powder preparations.

We prepared mixtures of basalt with various iron oxides including ferrihydrite, hematite, goethite, and akaganeite at a constant 1:2 weight ratio. For ferrihydrite and hematite specifically, we explored additional mixing ratios ranging from 1:99 to 1:1 with basalt. The specific mixing ratios used in this study are given in Supplementary Table 1. To investigate the 3-μm band shape and potential presence of sulfates in Martian dust, we created mixtures of ferrihydrite, basalt, and Mg-sulfate. While anhydrous $MgSO_4$ from Sigma-Aldrich Co. was used, spectral analysis revealed multiple hydration features characteristic of hexahydrite ($MgSO_4 \cdot 6H_2O$). These hydration features reflect structurally bound water molecules within the crystal structure, rather than adsorbed water, indicating hydration has occurred. All sample powders were homogenized through gentle disaggregation and mixing using a pestle and mortar.

## VNIR and FTIR spectroscopy

The spectra analyzed in this work were acquired under ambient conditions for all samples. The VNIR range of each spectrum was acquired with MoHIS[110], which is a high-resolution imaging spectrometer that covers from 0.4 to 2.4 μm. The near infrared (NIR) part up to 4 μm was acquired at IPAG Grenoble with a Brucker Vertex 70 V FTIR spectrometer equipped with a biconical reflectance kit A513/QA (i = 0°, e = 30°). To obtain a complete spectrum of each iron oxide phase, we spliced the VNIR and FTIR datasets at around 1.8 μm, resulting in the composite spectra shown in this work. Spectra of ferrihydrite-basalt mixtures in the VNIR and NIR range were acquired first at IPAG using SHADOWS spectro-gonio-radiometer. A more in-depth analysis spanning also other (oxy)hydroxide mixtures such as akaganeite, schwertmannite and goethite was performed at Brown University's NASA RELAB facility. In addition, 3 μm observations of the sulfate-bearing mixtures were also acquired at RELAB. Measurements were acquired in dry air to remove potential atmospheric effects. The instruments used were the UV-Vis-NIR Bidirectional Spectrometer and the Thermo Nexus 870 FT-IR Spectrometer. The spectra were spliced in a similar way to provide high resolution spectra up to 4 μm. RELAB's nominal resolution is 1.5, 3, and 6 nm for the gratings 1200, 600, and 300, covering 300-880, 860-1800, and 1780-2600 nm in the standard operation.

## Root mean square error (RMSE) calculation

To compare the goodness of fit of our lab mixture VNIR reflectance spectra and the VNIR reflectance spectra acquired by orbiters and rovers, we employed a root mean square error calculation. For each observed spectrum $S_{obs}$, the original spectrum data $S_{obs}$, were retrieved and normalized at 800 nm. The observed normalization value $N_{obs, 800nm}$ was obtained from the data at 800 nm. The normalized observed spectrum $S_{obs, norm}$ was calculated by dividing the original observed spectrum by its normalization value:

$$S_{obs, norm} = \frac{S_{obs}}{N_{obs, 800nm}} \qquad (1)$$

The root mean square error (RMSE) was then calculated by comparing the normalized observed spectrum $S_{obs, norm}$ to the laboratory normalized spectrum $S_{lab}$ in the same way. This involved computing the differences $\Delta S$ between the two spectra, squaring these differences, averaging them, and finally taking the square root of the mean value:

$$S = S_{lab, norm} - S_{obs, norm} \qquad (2)$$

$$RMSE = \sqrt{\frac{1}{n}\sum_{i=1}^{n}(\Delta S_i)^2} \qquad (3)$$

where $n$ is the number of data points in the spectra. Data gaps in the CRISM and OMEGA datasets were linearly interpolated.

## CaSSIS photometry and lab spectrogoniometry

We analyzed approximately 200 unique observations from CaSSIS, utilizing the latest absolute and radiometric calibration[111,112]. Previous studies have demonstrated the CaSSIS capability to detect phase angle-dependent phenomena related to surface roughness and particle size properties[113], as well as ices under low signal-to-noise ratio conditions[114], underscoring the instrument's great performance and suitability for our spectral analysis of Martian dust.

To collect the individual dust spectra from each CaSSIS cube, we used the Environment for Visualizing Images (ENVI) software to draw regions of interest (ROIs) over relatively flat, spectrally homogenous and stable surfaces. Each spectrum was the result of averaging 100–1000 pixels in size (at 4.6 m/px scale). Images obviously contaminated by clouds or other atmospheric phenomena were discarded. Following past analyses[115] we also excluded observations above 70° phase because of increasingly dominant aerosol effects. Additionally, for each CaSSIS observation we modeled optical depth (τ) values, using the climatological assimilation of dust opacities derived from the Mars Climate Sounder (MCS) experiment[116]. This model serves as a proxy for the atmospheric conditions during the time of image acquisition and provides confidence when interpreting surface reflectance trends.

Phase curve ratios of ferrihydrite were acquired using PHysikalisches Institut Radiometric Experiment-2 (PHIRE-2[117]), which is a lab spectrogoniometer at the University of Bern. In order to mimic CaSSIS observations, PHIRE-2 was set to observe only in the principal plane and in the phase angle range of 5–70° (when i = 0°, phase = emission). PHIRE-2 was also equipped with the flight spare CaSSIS filters (NIR, RED, PAN & BLU), which allows direct spectral comparisons between the lab and orbital data. PHIRE-2 reflectance data was radiometrically calibrated using a standard diffuse reflector (Spectralon™) and outputted in units of REFF.

Before each measurement, the samples were flattened using a smooth metal spatula[118,119]. After flattening, each surface was sprinkled with the remaining sample material using very fine nylon sieves of various mesh sizes (SEFAR NITEX®). The sprinkling procedure was performed to simulate the surface texture effects of dust resulting from airfall settling.

## Ferrihydrite dehydration experiment

The Mini-ME vacuum chamber in the University of Winnipeg was used to simulate Mars surface conditions (e.g. ref. 120). A commercial-grade, dry $CO_2$ atmosphere was maintained by allowing a constant flow through the chamber, unless otherwise noted below. The powders were packed in aluminum wells that are held under a 9.4-mm-thick sapphire window that allows both UV-irradiance and reflectance spectroscopy.

The ferrihydrite-basalt samples were initially exposed to a ~5–7 mbar $CO_2$ atmosphere at 22 °C, and reflectance spectra were measured after 10 minutes without irradiation. Following this, the

samples were irradiated with a Heraeus DO802 deuterium lamp, with spectral radiance assumed to be identical to that of the DO902 measured by ref. [121], which has near-identical specifications. The methods used here were previously used to quantify the stability of the MastCam-Z calibration targets when exposed to UV-irradiation[122]. A solarmeter 8.0 UVC radiometer quantified the irradiance of the DO802 at 50 mm from the aperture through a 9.4 mm sapphire window, yielding 0.25 W m$^{-2}$. The radiance spectra were normalized to the measured irradiance at 258 nm and adjusted for sapphire window transmittance. Dual-pass transmittance measurements using a halon reflectance standard showed no significant atmospheric absorption (180–2500 nm) between air and 400 Pa $CO_2$, likely due to the short 1-2 mm path length, so atmospheric absorption was ignored. The samples were subjected to both vacuum and UV irradiation for 678 h until UV bulb failure, with a final measurement taken at 966 hours under vacuum conditions only. All spectral measurements were conducted at room temperature (22 °C).

Reflectance spectra over the 350- to 2500-nm range were measured with an Analytical Spectral Devices FieldSpec spectrometer with a spectral resolution of between 2 and 7 nm and spectral sampling interval of 1.4 nm for the 350–1000 nm interval and 2 nm for the 1000- to 2500-nm interval. The data are internally resampled by the instrument to output data ultimately at 1-nm intervals with a cubic spline interpolation. Spectra were measured at a viewing geometry of i = 40° and e = 0° with incident light being provided by an in-house quartz-tungsten-halogen collimated light source (<1.5° divergence). Sample spectra were measured relative to halon that is held in an aluminum well inside the vacuum chamber, which was calibrated to a Fluorilon® 99% diffuse reflectance standard and corrected for its absorption properties according to the ASTM-E1331 standard method. For the spectral data acquisition, 500 spectra of the dark current, standard, and sample were acquired and averaged, to provide high signal-to-noise data. Reflectance spectra over the 2500–5000 nm range were collected at Mars pressures before irradiation and after the experiment was completed with a Designs & Prototypes Model 102 F emission spectrophotometer. Samples were illuminated with the same light source used for VNIR reflectance, but at a much lower power to avoid saturation of the interferograms. The sample spectra were collected relative to an aluminum powder held inside the vacuum chamber, and then were normalized relative to the data collected with the VNIR spectrometer at 2500 nm. Self-emission of the spectrophotometer was corrected by measuring a blackbody at 10 °C and 50 °C, and the correction was applied to all collected spectra. Reference spectra and sample spectra were acquired with the same viewing geometry of i = 40° and e = 0°, and 100 spectra were averaged for each collection.

X-ray diffraction patterns were acquired with a continuous 2-theta scan from 5 to 80° 2θ on a Bruker D8 Advance. A Bragg-Brentano goniometer with a two theta/theta setup was equipped with a 2.5° incident Soller slit, a 1.0 mm divergence slit, a 2.0 mm scatter slit, a 1.0 mm receiving slit, a curved secondary graphite monochromator, and a scintillation counter collecting at an increment of 0.02° and integration time of 1 s per step. The line focus Co X-ray tube was operated at 40 kV and 40 mA, using a take-off angle of 6°. Samples that had not been exposed to Mars-like atmospheric pressure and radiation were front-loaded into an X-ray amorphous sample cup. Samples that had been exposed to Mars-like conditions were sprinkled onto a zero-background plate due to limited sample volume. Phase identification was made using the Bruker EVA software package linked to the International Center for Diffraction Data Powder Diffraction File (ICDD PDF-2) database.

## Data availability

Source data are provided with this paper. In addition, the spectral data acquired at RELAB will be deposited on the PDS Geoscience node https://pds-speclib.rsl.wustl.edu/ and https://sites.brown.edu/relab/ relab-spectral-database/. Data acquired at University of Bern can be found under this reference: Valantinas, Adomas; Ottersberg, Rafael; Pommerol, Antoine (2023), Vis-NIR reflectance spectra of several powdered iron oxides, SSHADE/BYPASS (OSUG Data Center), Dataset/Spectral Data, https://doi.org/10.26302/SSHADE/EXPERIMENT_OP_20250122_001. Data acquired at IPAG (University of Grenoble) can be found under this reference: Valantinas, Adomas; Beck, Pierre; Poch, Olivier (2023): NIR reflectance spectra of several powdered iron oxides. SSHADE/GhoSST (OSUG Data Center). Dataset/Spectral Data. https://doi.org/10.26302/SSHADE/EXPERIMENT_LB_20250114_001. The CaSSIS images are available online at https://observations.cassis.unibe.ch/ and the Planetary Science Archive: https://psa.esa.int/.

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

## Acknowledgements

A.V. acknowledges R. Milliken for access to RELAB (multiuser facility supported by NASA PSEF program) N. Chatterjee (MIT Electron Microprobe Facility), D. Khan and N. Kopacz for field site identification at Block Island and Azores respectively, and P. O. Huber for graphic design assistance. CaSSIS is a project of the University of Bern and funded through the Swiss Space Office via ESA's PRODEX programme. The instrument hardware development was also supported by the Italian Space Agency (ASI) (ASI-INAF agreement no.2020-17-HH.0), INAF/Astronomical Observatory of Padova, and the Space Research Center (CBK) in Warsaw. Support from SGF (Budapest), the University of Arizona (Lunar and Planetary Lab.) and NASA are also gratefully acknowledged. Operations support from the UK Space Agency under grant ST/R003025/1 is also acknowledged. The experimental work of A.P. and R.O. has been carried out within the framework of the NCCR PlanetS supported by the Swiss National Science Foundation under grant 51NF40_205606. E.A.C. Was funded through grants from the Canada Foundation for Innovation and Research Manitoba (CFI-MRIF 1504), the Natural Sciences and Engineering Research Council of Canada (RGPIN-2023-03413) and the Canadian Space Agency (22EXPCOI4). N.M. participation has been granted by the Centre National d'Etudes Spatiales (CNES).

## Author contributions

A.V. conceived the study, analyzed orbital and laboratory data, interpreted the results, collected the natural ferrihydrite samples, performed kinetic calculations and wrote the manuscript. J.F.M. supervised the project, contributed to the interpretation of results, and revised the manuscript. V.C. synthesized and provided most of the iron (oxy)hydroxide samples, and constructed Pourbaix diagrams through thermodynamic modeling. T.H., P.B., A.S. and R.O. acquired spectral data. O.P. assisted with using the planetary grinder. D.M.A. conducted the Mars chamber experiments. M.R.P. and G.L.V. performed optical depth simulations. J.F.M., A.P., J.L.B., V.C., N.M., E.A.C., K.R. and G.L.V. contributed to discussions and assisted with data interpretation. A.P., P.B., J.L.B., J.F.M., N.M., O.P., V.C., D.M.A., E.A.C., S.P.-L. and A.S. contributed to writing. N.T. acquired funding.

## Competing interests

The authors declare no competing interests.
