## [Peer Review File · Nature Communications]

Detection of ferrihydrite in Martian red dust records ancient cold and wet conditions on Mars

Corresponding Author: Dr Adomas Valantinas

Version 0:

Reviewer comments:

Reviewer #1

(Remarks to the Author)

Review report of the manuscript "Why Mars is Red - Ferrihydrite Detection in Martian Dust and Implications for Early Mars" submitted to Nature Communications, NCOMMS-24-49227.

Dear Authors,

Thank you for the opportunity to review your manuscript. I enjoyed reading the manuscript and seeing the number of evidence that sub-micron ferrihydrite explains the observations. Overall, the manuscript proposes an interesting hypothesis supported by reasonable data and thoroughly describes the implications of the finding. Thus, I consider that there is no need for major corrections. Several minor comments are listed below, line by line.

Good luck and all the best

Minor comments

Line 247: Figure 7 appears before Figure 6. Please reconsider the order.

Line 318: Could you add an explanation or reference why the basalt may inhibit the transformation of ferrihydrite?

Line 357: Could you discuss the interpretation of the sub-micron size nature of ferrihydrite for the origin of the ferrihydrite? I suppose it is also an important constraint suggested by this study.

Line 371: A space after (atmospheric photolysis) is lacking.

Line 377: Could you add a brief explanation of why such a reaction occurs?

Line 494: Please reconsider the notation and explanation of Ferrihydrite-2L (2-line) in Fig. 8, as it is not reader-friendly.

Line 518: Information on the origin of goethite seems to be lacking.

Line 519: Although the reference is cited (Schwertmann and Cornell, 2000), I suggest briefly explaining the synthesis procedure, especially for ferrihydrite, as it is the key material of this study and would be of interest to readers.

Line 524: I suggest explicitly explaining that, from here, the description of the sample preparation begins, rather than just "First..." and "Second...".

Line 529: maghemeite => maghemite

Line 530: Please add spaces between < and numbers.

Line 530: Please reconsider the flow of paragraphs. For example, please state that this paragraph describes the preparation procedure for "7 samples" out of the 9 mentioned above paragraph and that the preparation procedure for others, including ferrihydrite, is written in the next paragraph so that readers can understand easily.

Line 531: Please add a description of the origin and nature of the basalt used in this study.

Reviewer #2

(Remarks to the Author)

Mars is a red planet. Previously, the red color on the surface of Mars was thought to be due to the ubiquitous presence of the anhydrous iron oxide hematite, which is associated with dusts on the planet. However, this study shows that the ubiquitous iron oxide on Mars is not hematite, but ferrihydrite. I think the finding is quite novel and has major implications for the past aqueous environment on Mars, which is worth publishing in the journal Nature Communication.

I agree with the authors that significant ferrihydrite with basalt can explain the red color of the dust on Mars. However, I am still concerned that (1) the presence of hematite rather than ferrihydrite is not completely ruled out (2) the ferrihydrite used in the study is the reasonable standard material for comparison with the spectra of Martian dust. In order to reach the authors' conclusion, these concerns need to be addressed.

Regarding the first point, the author made the systematic study using ferrihydrite and basalt to reproduce the spectra of the Mars spectra. However, they only showed the comparison with the 1:2 hematite-basalt mixture in Figure 2. Is it really impossible to reproduce the Mars spectra with hematite not ferrihydrite? As the authors also mentioned, the MSL measured the chemical compositions and mineralogy of the Rocknest site (dust sample) in Gale. The quantitative analysis of the XRD pattern suggests the presence of 1% hematite in the sample. Does the 1% hematite never explain the color of the Martian dust?

Regarding the second point, the color (or visible spectrum) of ferrihydrite must vary with conditions. The authors also mentioned that the spectrum of natural ferrihydrite is different from the synthesized material. Possible ranges of spectra of ferrihydrite are characteristic of other iron oxides, especially for hematite.

Minor comments:

Line 123-128: The motivation and purpose of this study should be written in the end of the introduction.

Line 126: What does it mean "stable"? What is the definition of the term of "stable" in this study?

Line 173: The dust sample from the Rocknest site contains hematite. Ferrihydrite is a metastable phase and changes to hematite and/or goethite. I think the mixture of ferrihydrite and hematite is a reasonable scenario for Martian dust, although the contribution of ferrihydrite may be much higher than hematite. The authors examined the inclusion of goethite with ferrihydrite (Supplementary Figure 8). The inclusion of hematite with ferrihydrite should be investigated, which is related to the main comment (1).

Line 237-: However, the periods that ferrihydrite survives must be 8-9 orders of magnitude longer than the experimental duration. These experiments may not provide evidence that ferrihydrite is stable on Mars. This comment is related to the second comment in the Minor Comments.

Line 247: Figure 7 should be Figure 6.

Reviewer #3

(Remarks to the Author)

The goal of this manuscript is to constrain the mineral assemblage of the martian dust primarily using visible/near-infrared spectroscopy. This manuscript presents visible/near-infrared reflectance data of Mars-relevant iron-bearing phases measured under modern martian atmospheric conditions (e.g., composition, radiation, pressure) and compares the laboratory spectra to martian orbital and in-situ spectra. The manuscript finds that a mixture of ferrihydrite and basalt, along with magnesium sulfate and goethite, best match the visible/near-infrared spectra from Mars. The manuscript concludes that the presence of ferrihydrite in martian dust can help constrain the early aqueous history of Mars and that the ferrihydrite formed just before the martian surface became hyperarid.

The topic of the manuscript will be of broad interest to the Mars science community because it addresses a long-standing question: What is the martian dust made of? The manuscript will also be of broad interest to the planetary spectroscopy community because it presents visible/near-infrared reflectance data of a variety of iron-bearing phases measured under Mars-like conditions. The methods, results, and conclusions are generally sound, but I think the manuscript should be rearranged a bit to highlight the important results and conclusions more clearly. For example, some data in the supplementary information should be moved to the main text, and a brief description of the methodology should be moved to the main text. I have a few substantial comments below, followed by minor line-by-line comments (most of which are suggested grammatical changes). If the authors have any questions about my review, they should feel free to email me, Liz Rampe, at elizabeth.b.rampe@nasa.gov.

Broad comments/suggestions

(1) Abstract: I suggest mentioning the results of the irradiation experiments in the abstract. The spectral measurements of different Fe phases under martian conditions (e.g., UV radiation, CO₂-rich atmosphere) will be important data for the planetary community to interpret spectra collected from a variety of planetary bodies.

(2) Introduction:

a. The introduction to Fe-oxides/oxyhydroxides in the martian dust is thorough, but I suggest adding more about how ferrihydrite and other Fe phases are used to interpret the past martian climate (e.g., under what conditions does ferrihydrite form and transform into other phases?).

b. CaSSIS data are not described in the introduction. I suggest introducing these data, what they're used for, and how they've been previously interpreted to justify the analyses of CaSSIS data in the manuscript.

c. Add a paragraph between the Introduction and Results sections that briefly describes the mixtures that were created so that the reader knows what's coming in the results.

(3) Methods:

a. Describe the composition of the basalt in the mineral mixtures (i.e., mineral assemblage, geochemical composition) to help the reader better interpret the spectra.

b. Further explain why Fe-oxides aside from hematite were not finely ground for spectral measurements.

c. Add information about the natural samples that were measured (e.g., sites, collection procedures, previous publications) in the supplementary information or methods.

(4) Results: Spectra of basalt-ferrihydrite-Mg sulfate mixtures show a good match to the 3-micron feature on Mars, but these

data are in the supplementary information and sulfates are not mentioned in the introduction. I suggest briefly discussing the importance of sulfates on Mars in the introduction and moving the basalt-ferrihydrate-Mg sulfate mixture methods and results to the main body of the manuscript. Explain how Mg sulfate fits into the timing of the formation of ferrihydrate if a ferrihydrate-basalt-Mg sulfate-goethite mixture best explains the vis/NIR data.

(5) Discussion: The discussion of how ferrihydrate formed on Mars could be better developed. There are some seemingly contradictory statements in the discussion about the timing of the formation of ferrihydrate. Did ferrihydrate form early or late in Mars' history? Did it form within a rock and then become physically eroded to get incorporated into the dust? Or did it form from water-rock interaction of unconsolidated eolian materials? If it originally formed in a rock then was physically eroded, why aren't there other secondary phases common on Mars in the martian dust (e.g., smectite)?

Line-by-line comments

Main text

Abstract: define all acronyms in the abstract.

The manuscript uses "um" and "micron" interchangeably. I suggest using one term and being consistent throughout the manuscript. I also suggest changing ampersands to "and".

Line 28: add an alpha symbol before "Fe₂O₃"

Line 31: clarify that the mineralogy of the surface dust is unresolved from MSL data (e.g., "..., but the minerals present in the dust remains unresolved).

Line 45: This introductory sentence indicates that the Fe-oxide assemblage only tells us about ancient climates. From your measurements of Fe-oxides/oxyhydroxides under modern martian conditions, the secondary Fe assemblage can also tell us about modern climate and how long those conditions may have persisted.

Line 46: add a comma after "Earth"

Line 48: add "typically" after "The" because the martian surface isn't red everywhere.

Line 55: Hyphenate "near-infrared"

Line 56: Define the acronym "ESA"

Line 58: Add a space between ")"("

Line 69: Hyphenate "surface-to-volume"

Line 76: Explain what bond vibrations cause the 1.4 and 1.9 um bands

Line 77: Add a comma after "e.g."

Line 79: Specify what "coarse-grained" means (e.g., what is the crystallite size?)

Line 81: Add "e.g.," before "Morris et al."

Line 82: add a comma after "(Fe₃₊/FeT)" and delete "more"

Line 84: Define "microcrystal" size range

Line 88: Change "while" to "whereas"

Line 92: Change "spectral" to "spectra"

Line 107: hyphenate "dust-covered" and change "dune" to "shadow"

Line 110: The abundance of Fe oxides in the amorphous component in Rocknest is unconstrained. We have calculated the abundance of Fe in the amorphous component using CheMin mineral abundances and APXS elemental abundances, but we don't know the speciation of the Fe (e.g., oxides, oxyhydroxides, sulfates).

Line 116: Change "at" to "in the" and change "soils" to "soil"

Line 121: Change "crystalline phyllosilicate minerals" to "hydrated or hydroxylated minerals"

Line 128: After reading the introduction, I suggest adding a paragraph on the methods before moving into the results to help prepare the reader for what data they will see in the results. For example, I was surprised to learn that you made measurements of mineral mixtures and not just individual minerals.

Line 133: clarify whether the FH-basalt powder was measured under martian conditions

Line 148: Delete the comma between CRISM and OMEGA

Line 168: Remove the accent in "akaganeite"

Line 181: Delete "VIS"

Line 182: Add a comma after "slope"

Line 186: Clarify what the amount of oxidized iron is smaller than

Line 218: Is it possible that the shoulder feature at ~600 nm could be from chemical substitutions or impurities in ferrihydrate (e.g., Si)?

Line 231: Add a comma after "e.g."

Line 231: I suggest changing the angle to Angstroms.

Line 234: At the end of the sentence, consider adding a clarifying clause like: "...may contain small amounts of other (oxy)hydroxides such as goethite, which contain a ~600 nm spectral feature."

Line 234: Add a comma after "basalt"

Line 241: This is the first mention of mixtures with sulfate. I suggest adding some text about sulfates in the introduction to motivate the measurement of mixtures with sulfate.

Lines 249 & 250: hyphenate "3.0-um" and "1.9-um"

Line 280: hyphenate "iron-oxide"

Line 284: Although there is very little H₂O in the martian atmosphere, the relative humidity can be very high sometimes, right? Consider changing "extremely low humidity" to something like "extremely low water vapor concentration".

Line 289: The ferrihydrate structure did not change over the course of the 40-day experiments presented here. The ferrihydrate in the martian dust has been exposed to these conditions for many thousands/millions of years. Would you expect the ferrihydrate structure to change over those much longer timescales?

Line 294: Add the calculations using the Arrhenius equation to the supplementary information

Line 301: Add "e.g.," before Fairén et al

Line 305: Add "e.g.," before Klingelhofer and consider adding references for the jarosite detection by CheMin (e.g., Rampe et

al., 2017; Morrison et al., 2024)

Rampe, E. B., Ming, D. W., Blake, D. F., Bristow, T. F., Chipera, S. J., Grotzinger, J. P., et al. (2017). Mineralogy of an ancient lacustrine mudstone succession from the Murray formation, Gale crater, Mars. *Earth and Planetary Science Letters*, 471, 172-185.

Morrison, S. M., Blake, D. F., Bristow, T. F., Castle, N., Chipera, S. J., Craig, P. I., et al. (2024). Expanded insights into Martian mineralogy: Updated analysis of Gale crater's mineral composition via CheMin crystal chemical investigations. *Minerals*, 14(8), 773.

Line 309: Add "e.g.," before Bishop & Murad

Line 318: Please explain how basalt may inhibit the transformation of ferrihydrite to a more crystalline phase.

Line 323: Add "e.g.," before Childs & Parfitt

Line 326: Add references that show ferrihydrite is more prevalent than allophane and imogolite in Fe-rich volcanic materials

Line 328: In the discussion, further explain why the presence of ferrihydrite provides insight into early Mars history and not more modern environments. Consider defining "early history" (e.g., number of billions of years ago).

Line 337: Add "e.g.," before Carr & Head

Line 350: The statement that ferrihydrite may have formed from the last widespread aqueous episode seems to contradict the statement that ferrihydrite formed during early Mars history.

Line 360: Add a comma after "solutions"

Line 366: Add "e.g.," before Burns

Line 385: Change "+2" to "2+"

Line 398: Capitalize "Mars Sample Return"

Line 398: Change "bring back regolith samples" to "deliver regolith samples to Earth"

Line 400: Consider elaborating on what we can learn about ferrihydrite and how it formed in the returned samples. What types of measurements should be performed on the regolith? Could the short-range order of ferrihydrite tell us if ferrihydrite is pure or beginning to transform to another, more crystalline Fe phase? Could the physical relationship between ferrihydrite and other phases tell us what has been altered to form ferrihydrite?

Line 403: Hyphenate "light-toned"

Figure 2b: Include the spectrum of basalt for comparison to the mixtures. Show spectra of mixtures with other Fe phases (e.g., akaganeite) either here or in the supplementary information.

Figure 5c: Label ferrihydrite peak positions in angstroms. Show the search-match results in the supplementary information.

Figure 5d: Show the LIBS spectrum for comparison to the shoulder feature in Mars data.

Figure 5: Could impurities in ferrihydrite contribute to the shoulder at ~650 nm?

Figure 6: Explain the reason for the difference in the curves at high phase angles (e.g., there is a roll off in Mars data but not in the lab data)

Figure 7a: Explain the reason for the shift in the 1.4-um band

Line 488: Hyphenate "background-removed"

Line 491: Hyphenate 1.9-um

Figure 8: Explain how the figure was made in the methods section. Also explain what "Ferrihydrite-2L" means.

Line 499: Move the RMSE calculation section after the VNIR and FTIR spectroscopy section.

Line 500: Clarify that the spectra are VNIR reflectance spectra (e.g., "To compare the goodness of fit of our lab mixture VNIR reflectance spectra and the VNIR reflectance spectra acquired by orbiters...")

Line 501: Add a comma after "rovers"

Line 518: Add a comma after "goethite"

Line 520: Clarify whether iron oxide powders were analyzed via XRD before the experiments to confirm purity and that the minerals were not altered over the course of 14 years.

Line 524: Add a comma after "samples"

Line 524: Specify whether sieving was done dry or wet

Line 526: Change "sample" to "samples"

Line 528: Fix the spelling of goethite

Line 531: Add a comma after "size"

Line 531: Explain the source of the basalt in the mixtures

Line 540: State the temperature under which the ethanol was evaporated

Line 545: Explicitly state the proportions of minerals and basalt in the mixtures in a table in the methods or supplementary information

Line 554: I suggest rearranging the sentence to "Spectra of ferrihydrite-basalt mixtures in the VNIR and NIR range were acquired..."

Line 556: Specify which phases and mixtures were measured at RELAB

Line 559: Delete "spectral"

Line 560: Define the spectral resolution of the instruments.

Line 576: Define "MCS"

Line 576: Define what "This" refers to (e.g., "This model...")

Line 597: Hyphenate "mm-thick"

Line 599: State for how long the samples were exposed to martian atmospheric and radiation conditions before spectral measurements. Also state the temperature of the measurements.

Line 599: Add a comma after "atmosphere"

Line 601: Delete "was used"

Line 623: The sentence that starts on line 623 doesn't read correctly to me. I think "data." should be added after "signal-to-noise", and "reflectance" should be the start of a new sentence.

Line 633: Add a comma after "0 deg"

Line 640: Be more descriptive of "pre-" and "post-exposed". For example, "Samples that had not been exposed to Mars-like

atmospheric pressure and radiation were front-loaded..."

Line 641: Change "sample was" to "samples were"

Supplementary information

Line 67: Add a comma after "mixtures"

Line 71: Change "comparison" to "compared"

Line 79: Fix the spelling of goethite

Line 88: Explain why magnetite is not shown in Supplementary Fig. 5.

Supplementary Fig. 6: Explain the source of the K in the synthetic ferrihydrite and whether it affects the ferrihydrite spectral properties.

Line 101: Add a comma after "hematite"

Supplementary Fig. 8: Show the martian spectrum for comparison to laboratory spectra.

Line 110: Add a comma after "basalt"

Line 112: Change "a goethite" to "as goethite"

Line 119: Describe and justify in the methods the type of Mg sulfate that was used in the mixtures (e.g., epsomite)

Supplementary Fig. 10: Change c and d to a and b

Version 1:

Reviewer comments:

Reviewer #2

(Remarks to the Author)

The authors fully address my concerns. I am happy to recommend acceptance of the manuscript to the journal Nature Communication.

Reviewer #3

(Remarks to the Author)

Thank you to the authors for their detailed and thoughtful responses to my comments. I believe the authors have sufficiently addressed my comments, and I recommend the manuscript be published in Nature Communications without further edits.

General reply to the reviewers

Thank you for your careful and thorough reviews that have significantly improved our manuscript. We would like to highlight several key points addressed in our revision.

First, regarding the crucial question of ferrihydrite stability on present day Mars raised by two reviewers. We have substantially expanded our discussion of this topic, integrating both our experimental observations (14-year stability under Earth conditions and 40-day Mars-simulation experiments) with recent theoretical calculations by Sassi & Rosso (2022). Using their model we demonstrate that at Mars' average temperature (210 K), the kinetic barriers to iron self-diffusion effectively prevent solid-state transformation, with estimated timescales of 10^{16} years far exceeding the age of the solar system.

In addition, several moderate changes have been implemented in response to your comments:

1. We conducted an electron microprobe analysis of our synthetic ferrihydrite at MIT's Electron Microprobe Facility, replacing our previous XRF data, which was erroneous. This analysis revealed that in our preferred 1:2 and 1:3 ferrihydrite-to-basalt mixtures there is about 20-30 wt. % poorly crystalline FeO, thereby increasing our previous estimation.
2. Following Reviewer #3's suggestion, we performed complete XRD characterization of all iron oxide-hydroxide phases. This analysis revealed that our primary "schwertmannite" (oxyhydroxysulfate) sample was actually akaganeite (hydroxide), likely due to a labeling error during sample preparation 14 years ago. We have corrected this throughout the manuscript and figures, particularly in Figure 2b which now shows the "Akaganeite-basalt mix" spectrum. We also included a second, verified schwertmannite sample (containing minor natrojarosite) to maintain comprehensive coverage of iron oxide phases.
3. We expanded the spectral analysis to include additional mixing ratios, particularly addressing Reviewer #2's concerns about hematite content. These new results are presented in supplementary figures demonstrating that even small amounts of crystalline hematite would be spectrally detectable.
4. We clarified the manuscript's structure and improved our discussion of formation timing, focusing on the Hesperian period (~3 Ga). To illustrate the key concepts, we also included a new schematic Fig. 9. We also acknowledge that precise age determinations require additional constraints beyond our current scope.

The manuscript has been revised throughout to incorporate these changes along with numerous other suggested improvements. We thank the reviewers again for their valuable input. Please find our detailed responses to individual comments below.

Sincerely,

A. Valantinas and the team

REVIEWER COMMENTS

Reviewer #1 (Remarks to the Author):

Review report of the manuscript "Why Mars is Red - Ferrihydrite Detection in Martian Dust and Implications for Early Mars" submitted to Nature Communications, NCOMMS-24-49227.

Dear Authors,

Thank you for the opportunity to review your manuscript. I enjoyed reading the manuscript and seeing the number of evidence that sub-micron ferrihydrite explains the observations. Overall, the manuscript proposes an interesting hypothesis supported by reasonable data and thoroughly describes the implications of the finding. Thus, I consider that there is no need for major corrections. Several minor comments are listed below, line by line.

Good luck and all the best

Minor comments

Line 247: Figure 7 appears before Figure 6. Please reconsider the order.

Fixed.

Line 318: Could you add an explanation or reference why the basalt may inhibit the transformation of ferrihydrite?

This is a fair concern, thank you. We have decided to remove this statement from the manuscript.

Line 357: Could you discuss the interpretation of the sub-micron size nature of ferrihydrite for the origin of the ferrihydrite? I suppose it is also an important constraint suggested by this study.

Thank you for this suggestion. We have included a short text on the interpretation of the sub-micron size nature of ferrihydrite in the discussion section.

Line 371: A space after (atmospheric photolysis) is lacking.

This gets fixed after implementing Nature style citations.

Line 377: Could you add a brief explanation of why such a reaction occurs?

The oxidation reaction occurs because even though the thermodynamic equilibrium does not favor Fe³⁺ formation, the continuous escape of H₂ from the experimental system drives the reaction forward according to Le Chatelier's principle. As H₂ is removed, the system continues to produce more H₂ and Fe³⁺ to maintain equilibrium. The experiments demonstrated that iron oxidation could proceed without UV radiation or free oxygen, likely driven by this H₂ loss process. This information is added in the following sentence where we cite Chevrier et al. (2006).

Line 494: Please reconsider the notation and explanation of Ferrihydrite-2L (2-line) in Fig. 8, as it is not reader-friendly.

Fixed.

Line 518: Information on the origin of goethite seems to be lacking.

Fixed. This paragraph was reworded to improve clarity. Goethite sample is from the mine of En Bournegade, Tarn, France.

Line 519: Although the reference is cited (Schwertmann and Cornell, 2000), I suggest briefly explaining the synthesis procedure, especially for ferrihydrite, as it is the key material of this study and would be of interest to readers.

Thank you for this suggestion. We have added a detailed description of the ferrihydrite synthesis procedure to the Methods section, including the key steps of $\text{Fe}(\text{NO}_3)_3 \cdot 9\text{H}_2\text{O}$ dissolution, pH adjustment with KOH, and centrifugation protocols.

Line 524: I suggest explicitly explaining that, from here, the description of the sample preparation begins, rather than just "First..." and "Second..."

Fixed.

Line 529: maghemeite => maghemite

Fixed.

Line 530: Please add spaces between < and numbers.

Fixed.

Line 530: Please reconsider the flow of paragraphs. For example, please state that this paragraph describes the preparation procedure for "7 samples" out of the 9 mentioned above paragraph and that the preparation procedure for others, including ferrihydrite, is written in the next paragraph so that readers can understand easily.

Fixed.

Line 531: Please add a description of the origin and nature of the basalt used in this study.

Thank you for this suggestion. We have analyzed the mineralogical composition of the basalt sample using XRD and added these results to the Supplementary file. The analysis reveals several silicate mineral phases including augite, forsterite, magnetite, anorthite, nepheline and sanidine. Based on quantitative XRD analysis showing high abundances of sanidine (12.5 wt%) and nepheline (6.0 wt%), this sample shows characteristics of an alkaline basalt. While the exact sampling locality is unknown, we know the sample originates from Virginia, USA. This information has been added to the Methods section.

Reviewer #2 (Remarks to the Author):

Mars is a red planet. Previously, the red color on the surface of Mars was thought to be due to the ubiquitous presence of the anhydrous iron oxide hematite, which is associated with dusts on the planet. However, this study shows that the ubiquitous iron oxide on Mars is not hematite, but ferrihydrite. I think the finding is quite novel and has major implications for the past aqueous environment on Mars, which is worth publishing in the journal Nature Communication.

I agree with the authors that significant ferrihydrite with basalt can explain the red color of the dust on Mars. However, I am still concerned that (1) the presence of hematite rather than ferrihydrite is not completely ruled out (2) the ferrihydrite used in the study is the reasonable standard material for comparison with the spectra of Martian dust. In order to reach the authors' conclusion, these concerns need to be addressed.

Regarding the first point, the author made the systematic study using ferrihydrite and basalt to reproduce the spectra of the Mars spectra. However, they only showed the comparison with the 1:2 hematite-basalt mixture in Figure 2. Is it really impossible to reproduce the Mars spectra with hematite not ferrihydrite? As the authors also mentioned, the MSL measured the chemical compositions and mineralogy of the Rocknest site (dust sample) in Gale. The quantitative analysis of the XRD pattern suggests the presence of 1% hematite in the sample. Does the 1% hematite never explain the color of the Martian dust?

We appreciate this important question. Our team thoroughly investigated different hematite-to-basalt ratios and found that none could adequately reproduce the Martian dust spectrum. We have included a supplementary figure demonstrating this finding (see Fig. 1 here). As shown, even a 1:99 weight ratio produces a spectrum distinctly different from that of Martian dust, while a 1:9 ratio exhibits clear hematite spectral signatures (broad blue absorption at $\sim 0.5 \mu\text{m}$, shoulder at $0.6 \mu\text{m}$ and deep absorption at $0.85 \mu\text{m}$) in the VNIR range that are not observed in Martian spectra. These results strongly suggest that hematite cannot be the dominant iron oxide phase in Martian dust. This point was also added to the manuscript main text.

Figure 1. Hematite-basalt mix spectra of different weight ratios in comparison to Martian dust OMEGA spectrum. The 1:99 weight ratio produces a spectrum distinctly different from that of Martian dust, while a 1:9 ratio exhibits clear hematite spectral signatures (broad blue absorption at $\sim 0.5 \mu\text{m}$, shoulder

at 0.6 μm and deep absorption at 0.85 μm) in the VNIR range that are not observed in Martian spectra. These results strongly suggest that hematite cannot be the dominant iron oxide phase in Martian dust. Spectra normalized at 1.0 μm .

Regarding the second point, the color (or visible spectrum) of ferrihydrite must vary with conditions. The authors also mentioned that the spectrum of natural ferrihydrite is different from the synthesized material. Possible ranges of spectra of ferrihydrite are characteristic of other iron oxides, especially for hematite.

Thank you for this important observation. While ferrihydrite's spectral properties show some variation depending on formation conditions, the fundamental spectral differences between iron oxide phases remain diagnostically distinct. Our comparative analysis of natural and synthetic ferrihydrites versus crystalline hematite shows clear spectroscopic differences (see Fig. 2 here). These differences arise from crystal structure variations, which are confirmed by distinct X-ray diffraction patterns for each phase (Supplementary Fig. 15). The XRD patterns provide independent validation of the mineralogical differences observed in our VNIR spectral analysis.

Figure 2. Natural and synthetic ferrihydrite comparison to hematite used in this study. Diagnostic features marked with arrows. For clarity spectra normalized at 740 nm.

Minor comments:

Line 123-128: The motivation and purpose of this study should be written in the end of the introduction.

Thank you for this suggestion. We have added the following content to the beginning of the last paragraph: "While MSL's measurements established that iron oxides comprise ~20 wt. % of the X-ray

amorphous component in Martian soils, the specific mineralogical nature of these phases has remained elusive.”

Line 126: What does it mean “stable”? What is the definition of the term of “stable” in this study?

We added a clarification saying “We show that ferrihydrite maintains its mineralogical structure and does not transform into other iron oxide phases when exposed to simulated present-day Martian conditions” and removed the word “stable”.

Line 173: The dust sample from the Rocknest site contains hematite. Ferrihydrite is a metastable phase and changes to hematite and/or goethite. I think the mixture of ferrihydrite and hematite is a reasonable scenario for Martian dust, although the contribution of ferrihydrite may be much higher than hematite. The authors examined the inclusion of goethite with ferrihydrite (Supplementary Figure 8). The inclusion of hematite with ferrihydrite should be investigated, which is related to the main comment (1).

Thank you for this insightful suggestion. We have investigated ferrihydrite-hematite-basalt mixtures and added two new figures to address this point:

- 1. Supplementary Fig. 9b demonstrates that while 10 wt% hematite produces identifiable spectral signatures, very small amounts of hematite (<1 wt%) in ferrihydrite-basalt mixtures are spectrally undetectable.**
- 2. Supplementary Fig. 9c provides a direct comparison between 1 wt% hematite and 1 wt% goethite in these mixtures, showing similar spectra with goethite exhibiting a slightly more pronounced shoulder feature.**

These results indicate that while we cannot exclude the presence of minor amounts of crystalline hematite through spectroscopy alone, any significant hematite content would be detectable in our spectral analysis.

Line 237-: However, the periods that ferrihydrite survives must be 8-9 orders of magnitude longer than the experimental duration. These experiments may not provide evidence that ferrihydrite is stable on Mars. This comment is related to the second comment in the Minor Comments.

Thank you for raising this important question about ferrihydrite's long-term stability on Mars. While our experimental timescales are indeed much shorter than geological periods, both theoretical calculations and empirical observations strongly support ferrihydrite's stability under Martian conditions. The key lies in understanding the temperature dependence of transformation kinetics. Recent calculations by Sassi & Rosso (2022) reveal that ferrihydrite transformation requires iron cation self-diffusion within the crystal lattice. This process has high activation energy barriers (0.57-2.47 eV per unit cell), making it extremely slow even at Earth temperatures (300 K). Under these conditions, calculated transformation times range from decades to millions of years - far longer than the months to years observed in natural settings where water-mediated processes dominate. When we extend these calculations to Mars' average temperature (210 K), the transformation becomes effectively impossible - timescales reach 10^{16} years for the fastest pathway and 10^{25} years for the slowest, far exceeding the age of the solar system itself (see Fig. 3 here).

Our experimental observations provide additional support for these predictions. We found that our 2-line ferrihydrite remained stable for 14 years under ambient lab conditions, and even aggressive

dehydration and exposure to present-day Martian conditions (UV, low pressure, CO₂ atmosphere) for 40 days had no effect on its crystallinity (e.g. main text Fig. 7).

This highlights an important distinction between Earth and Mars environments. On Earth, ferrihydrite typically transforms through either solid-state transformation at elevated temperatures (>400 K) or dissolution-precipitation reactions mediated by liquid water. On Mars, the combination of low temperatures and absence of liquid water effectively "freezes" both transformation pathways. While ferrihydrite remains thermodynamically metastable compared to goethite, the extreme suppression of iron self-diffusion at 210 K prevents the structural reorganization needed for crystallization. Therefore, what appears unstable under Earth conditions can persist indefinitely in the Martian environment.

Figure 3. Solid-state transformation timescales of ferrihydrite to goethite.

Line 247: Figure 7 should be Figure 6.

Fixed.

Reviewer #3 (Remarks to the Author):

The goal of this manuscript is to constrain the mineral assemblage of the martian dust primarily using visible/near-infrared spectroscopy. This manuscript presents visible/near-infrared reflectance data of Mars-relevant iron-bearing phases measured under modern martian atmospheric conditions (e.g., composition, radiation, pressure) and compares the laboratory spectra to martian orbital and in-situ spectra. The manuscript finds that a mixture of ferrihydrite and basalt, along with magnesium sulfate

and goethite, best match the visible/near-infrared spectra from Mars. The manuscript concludes that the presence of ferrihydrite in martian dust can help constrain the early aqueous history of Mars and that the ferrihydrite formed just before the martian surface became hyperarid.

The topic of the manuscript will be of broad interest to the Mars science community because it addresses a long-standing question: What is the martian dust made of? The manuscript will also be of broad interest to the planetary spectroscopy community because it presents visible/near-infrared reflectance data of a variety of iron-bearing phases measured under Mars-like conditions. The methods, results, and conclusions are generally sound, but I think the manuscript should be rearranged a bit to highlight the important results and conclusions more clearly. For example, some data in the supplementary information should be moved to the main text, and a brief description of the methodology should be moved to the main text. I have a few substantial comments below, followed by minor line-by-line comments (most of which are suggested grammatical changes). If the authors have any questions about my review, they should feel free to email me, Liz Rampe, at elizabeth.b.rampe@nasa.gov.

Broad comments/suggestions

(1) Abstract: I suggest mentioning the results of the irradiation experiments in the abstract. The spectral measurements of different Fe phases under martian conditions (e.g., UV radiation, CO₂-rich atmosphere) will be important data for the planetary community to interpret spectra collected from a variety of planetary bodies.

Thank you for the suggestion. Abstract updated.

(2) Introduction:

a. The introduction to Fe-oxides/oxyhydroxides in the martian dust is thorough, but I suggest adding more about how ferrihydrite and other Fe phases are used to interpret the past martian climate (e.g., under what conditions does ferrihydrite form and transform into other phases?).

Thank you for this suggestion. We added two sentences to address this in the first introduction paragraph (and additional references).

b. CaSSIS data are not described in the introduction. I suggest introducing these data, what they're used for, and how they've been previously interpreted to justify the analyses of CaSSIS data in the manuscript.

Thank you for this suggestion. We have added a brief introduction to CaSSIS in the last paragraph: "...using multi-angular observations from the Colour and Stereo Surface Imaging System (CaSSIS) onboard ESA's Trace Gas Orbiter, which, combined with laboratory spectrogoniometric measurements of ferrihydrite, suggests that dust particles are predominantly sub μm (<1 μm) in size." Due to Nature Communications' 5000-word limit for the main text, we have placed the detailed historical usage and examples of CaSSIS observations in the Methods section, which is more appropriate for the journal's style.

c. Add a paragraph between the Introduction and Results sections that briefly describes the mixtures that were created so that the reader knows what's coming in the results.

Thank you for this suggestion. While Nature Communications' format requires the Methods section at the end of the article, we have addressed this concern by mentioning our key experimental approach in the introduction's final paragraph: "Through systematic visible near-infrared (VNIR) spectral analysis, combining orbital and in-situ observations with laboratory studies of novel submicron ferrihydrite-basalt mixtures..." This previews the experimental work while maintaining the journal's required structure.

(3) Methods:

a. Describe the composition of the basalt in the mineral mixtures (i.e., mineral assemblage, geochemical composition) to help the reader better interpret the spectra.

Thank you for this suggestion. We have analyzed and added the basalt's mineralogical composition in Supplementary Fig. 13. XRD analysis reveals a mineral assemblage of augite, forsterite, magnetite, anorthite, nepheline (6.0 wt%) and sanidine (12.5 wt%). The high sanidine and nepheline content, combined with low SiO₂ (40.83%), indicates this is an alkaline basalt. This mineralogical assemblage suggests an evolved alkaline magmatic system rather than a typical tholeiitic basalt composition.

b. Further explain why Fe-oxides aside from hematite were not finely ground for spectral measurements.

We included this information in the methods section: "Given the extensive time required for the specialized grinding procedure and the primary focus on comparing ferrihydrite and hematite as potential major components of Martian dust, we prioritized these two phases for submicron size preparation. The other iron (oxy)hydroxides with particle sizes ranging from <40 μm to <10 μm were included to demonstrate their distinct spectral properties and rule out their dominance as the primary iron-bearing phase in Martian dust, which could be accomplished effectively with their standard powder preparations."

c. Add information about the natural samples that were measured (e.g., sites, collection procedures, previous publications) in the supplementary information or methods.

Thank you for this suggestion. We have expanded our sample descriptions and collection procedures in the Methods section. Note that none of these samples have been published before. Most of the iron oxide-hydroxide samples come from the University of Arkansas collection. The natural ferrihydrites were recently collected by the lead author.

(4) Results: Spectra of basalt-ferrihydrite-Mg sulfate mixtures show a good match to the 3-micron feature on Mars, but these data are in the supplementary information and sulfates are not mentioned in the introduction. I suggest briefly discussing the importance of sulfates on Mars in the introduction and moving the basalt-ferrihydrite-Mg sulfate mixture methods and results to the main body of the manuscript. Explain how Mg sulfate fits into the timing of the formation of ferrihydrite if a ferrihydrite-basalt-Mg sulfate-goethite mixture best explains the vis/NIR data.

Thank you for this suggestion. We agree that the 3-micron band spectral match is an important novel finding that deserves distinction in the main text. We have moved the ferrihydrite-basalt-Mg sulfate mixture spectral data from the supplementary information to the main text. In the introduction, we've added a brief mention of sulfates, noting their identification in rocks and dust in Gale Crater by MSL instruments, with appropriate citations. We've also included a concise discussion of the spectral

results and implications in the results section. However, given Nature Communications' 5000-word limit and our current manuscript length, we had to be selective in expanding the introduction, results and discussion. We focus on the clear ferrihydrite-basalt-sulfate spectral relationship at 3 microns, while noting that the potential minor goethite component (suggested by the 600 nm feature) requires further study to confirm. This allows us to highlight the key novel findings while maintaining manuscript concision. The sulfate discussion is kept brief but impactful, emphasizing the spectral evidence while acknowledging that detailed analysis of formation timing relationships would require more extensive treatment better suited for future work.

(5) Discussion: The discussion of how ferrihydrite formed on Mars could be better developed. There are some seemingly contradictory statements in the discussion about the timing of the formation of ferrihydrite. Did ferrihydrite form early or late in Mars' history? Did it form within a rock and then become physically eroded to get incorporated into the dust? Or did it form from water-rock interaction of unconsolidated eolian materials? If it originally formed in a rock then was physically eroded, why aren't there other secondary phases common on Mars in the martian dust (e.g., smectite)?

Thank you for this important observation. We have revised the discussion to remove potentially contradictory statements about timing and clarified our key findings and scope.

We note that the goal of this paper is to determine the iron oxide phase in the dust and not to identify the source of the dust. While our data suggest ferrihydrite formed when liquid water was present on Mars (perhaps 3 billion years ago during the Late Hesperian period), we leave precise age determinations and source mechanisms for future work.

Regarding the apparent absence of smectites in Martian dust - our observations align with terrestrial analogs where ferrihydrite commonly forms in volcanic settings through early alteration of basaltic materials. Under cold and wet conditions, the early alteration of basaltic glass primarily produces iron hydroxides with minor amounts of imogolite and kaolinite, rather than smectites which typically require prolonged weathering under warmer, wetter conditions. This is analogous to young Andosols on Earth where ferrihydrite and allophane dominate over smectites.

Line-by-line comments

Main text

Abstract: define all acronyms in the abstract.

As it currently stands the abstract is 230+ words long and Nature allows only 200. Therefore, we have decided to delete the acronyms.

The manuscript uses "um" and "micron" interchangeably. I suggest using one term and being consistent throughout the manuscript. I also suggest changing ampersands to "and".

Fixed.

Line 28: add an alpha symbol before "Fe2O3"

Fixed.

Line 31: clarify that the mineralogy of the surface dust is unresolved from MSL data (e.g., "..., but the minerals present in the dust remains unresolved).

Fixed.

Line 45: This introductory sentence indicates that the Fe-oxide assemblage only tells us about ancient climates. From your measurements of Fe-oxides/oxyhydroxides under modern martian conditions, the secondary Fe assemblage can also tell us about modern climate and how long those conditions may have persisted.

Included the words "past and present".

Line 46: add a comma after "Earth"

Fixed.

Line 48: add "typically" after "The" because the martian surface isn't red everywhere.

Included a specifier "dust-covered".

Line 55: Hyphenate "near-infrared"

Fixed.

Line 56: Define the acronym "ESA"

Fixed.

Line 58: Add a space between "){"

This goes away with Nature-style referencing.

Line 69: Hyphenate "surface-to-volume"

Fixed.

Line 76: Explain what bond vibrations cause the 1.4 and 1.9 um bands

Fixed.

Line 77: Add a comma after "e.g."

Fixed.

Line 79: Specify what "coarse-grained" means (e.g., what is the crystallite size?)

Specified that mm-sized hematite spherules (aka blueberries) were identified.

Line 81: Add "e.g.," before "Morris et al."

Fixed.

Line 82: add a comma after "(Fe³⁺/FeT)" and delete "more"

Fixed.

Line 84: Define “microcrystal” size range

Fixed. Added “nanocrystal (<10 nm)”.

Line 88: Change “while” to “whereas”

Fixed.

Line 92: Change “spectral” to “spectra”

Fixed.

Line 107: hyphenate “dust-covered” and change “dune” to “shadow”

Fixed.

Line 110: The abundance of Fe oxides in the amorphous component in Rocknest is unconstrained. We have calculated the abundance of Fe in the amorphous component using CheMin mineral abundances and APXS elemental abundances, but we don’t know the speciation of the Fe (e.g., oxides, oxyhydroxides, sulfates).

Updated the sentence to include this information.

Line 116: Change “at” to “in the” and change “soils” to “soil”

Fixed.

Line 121: Change “crystalline phyllosilicate minerals” to “hydrated or hydroxylated minerals”

Fixed.

Line 128: After reading the introduction, I suggest adding a paragraph on the methods before moving into the results to help prepare the reader for what data they will see in the results. For example, I was surprised to learn that you made measurements of mineral mixtures and not just individual minerals.

Thank you for this suggestion. Due to Nature Communications' strict word limit and format requirements, we cannot add a separate methods paragraph before the Results. However, we have addressed this by expanding the final introduction paragraph to preview our experimental approach: "Through systematic visible near-infrared (VNIR) spectral analysis, combining orbital and in-situ observations with laboratory studies of novel submicron ferrihydrite-basalt mixtures..." This provides readers with key methodological context while maintaining journal formatting requirements.

Line 133: clarify whether the FH-basalt powder was measured under martian conditions

Mentioned that this spectrum was acquired under ambient conditions. The reason being is to highlight that even under ambient conditions hydration features except the notable 3-micron band are not observed.

Line 148: Delete the comma between CRISM and OMEGA

Fixed.

Line 168: Remove the accent in “akaganeite”

Fixed.

Line 181: Delete "VIS"

Fixed.

Line 182: Add a comma after "slope"

Fixed.

Line 186: Clarify what the amount of oxidized iron is smaller than

Thank you for noting this unclear statement. We have removed this sentence about relative amounts of oxidized iron since it was ambiguous and potentially confusing. Instead, we focus on the subsequent, more concrete discussion of total FeO content, which provides clearer context for our findings.

Line 218: Is it possible that the shoulder feature at ~600 nm could be from chemical substitutions or impurities in ferrihydrite (e.g., Si)?

Thank you for this interesting suggestion. While ferrihydrite commonly contains impurities on Earth, we find it unlikely that the 600 nm feature is related to chemical substitutions for several reasons:

- 1. Previous studies of Si-rich ferrihydrites (Bishop & Murad, 2002) do not show this spectral shoulder.**
- 2. This spectral region is highly characteristic of iron oxide crystal field transitions, with several phases (goethite, hematite, akaganeite) showing absorption features around 650 nm that result in similar spectral shoulders (Scheinost et al. 1998; see Supplementary Fig. 4b).**

Line 231: Add a comma after "e.g."

Fixed.

Line 231: I suggest changing the angle to Angstroms.

Fixed.

Line 234: At the end of the sentence, consider adding a clarifying clause like: "...may contain small amounts of other (oxy)hydroxides such as goethite, which contain a ~600 nm spectral feature."

Fixed.

Line 234: Add a comma after "basalt"

Fixed.

Line 241: This is the first mention of mixtures with sulfate. I suggest adding some text about sulfates in the introduction to motivate the measurement of mixtures with sulfate.

We added a brief mention of sulfates in the introduction section. "ChemCam also detected sulfates in the rocks(Rapin et al., 2019) and dust(David et al., 2022) in Gale Crater indicating an ancient evaporitic and acidic environment"

Lines 249 and 250: hyphenate “3.0-um” and “1.9-um”

Fixed.

Line 280: hyphenate “iron-oxide”

Fixed.

Line 284: Although there is very little H₂O in the martian atmosphere, the relative humidity can be very high sometimes, right? Consider changing “extremely low humidity” to something like “extremely low water vapor concentration”.

Correct. We updated this sentence to include “extremely low water vapor content”.

Line 289: The ferrihydrite structure did not change over the course of the 40-day experiments presented here. The ferrihydrite in the martian dust has been exposed to these conditions for many thousands/millions of years. Would you expect the ferrihydrite structure to change over those much longer timescales?

(copy pasted from the response to rev #2)

Thank you for raising this important question about ferrihydrite's long-term stability on Mars. While our experimental timescales are indeed much shorter than geological periods, both theoretical calculations and empirical observations strongly support ferrihydrite's stability under Martian conditions. The key lies in understanding the temperature dependence of transformation kinetics. Recent calculations by Sassi & Rosso (2022) reveal that ferrihydrite transformation requires iron cation self-diffusion within the crystal lattice. This process has high activation energy barriers (0.57-2.47 eV per unit cell), making it extremely slow even at Earth temperatures (300K). Under these conditions, calculated transformation times range from decades to millions of years - far longer than the months to years observed in natural settings where water-mediated processes dominate. When we extend these calculations to Mars' average temperature (210K), the transformation becomes effectively impossible - timescales reach 10^{16} years for the fastest pathway and 10^{25} years for the slowest, far exceeding the age of the solar system itself (see Fig. 3 here).

Our experimental observations provide additional support for these predictions. We found that our 2-line ferrihydrite remained stable for 14 years under ambient lab conditions, and even aggressive dehydration and exposure to present-day Martian conditions (UV, low pressure, CO₂ atmosphere) for 40 days had no effect on its crystallinity (e.g. main text Fig. 7).

This highlights an important distinction between Earth and Mars environments. On Earth, ferrihydrite typically transforms through either solid-state transformation at elevated temperatures (>400K) or dissolution-precipitation reactions mediated by liquid water. On Mars, the combination of low temperatures and absence of liquid water effectively "freezes" both transformation pathways. While ferrihydrite remains thermodynamically metastable compared to goethite, the extreme suppression of iron self-diffusion at 210K prevents the structural reorganization needed for crystallization. Therefore, what appears unstable under Earth conditions can persist indefinitely in the Martian environment.

Line 294: Add the calculations using the Arrhenius equation to the supplementary information

Done.

Line 301: Add “e.g.,” before Fairen et al

Fixed.

Line 305: Add “e.g.,” before Klingelhofer and consider adding references for the jarosite detection by CheMin (e.g., Rampe et al., 2017; Morrison et al., 2024)

Rampe, E. B., Ming, D. W., Blake, D. F., Bristow, T. F., Chipera, S. J., Grotzinger, J. P., et al. (2017). Mineralogy of an ancient lacustrine mudstone succession from the Murray formation, Gale crater, Mars. *Earth and Planetary Science Letters*, 471, 172-185.

Morrison, S. M., Blake, D. F., Bristow, T. F., Castle, N., Chipera, S. J., Craig, P. I., et al. (2024). Expanded insights into Martian mineralogy: Updated analysis of Gale crater’s mineral composition via CheMin crystal chemical investigations. *Minerals*, 14(8), 773.

Fixed and added.

Line 309: Add “e.g.,” before Bishop and Murad

Fixed.

Line 318: Please explain how basalt may inhibit the transformation of ferrihydrite to a more crystalline phase.

We have removed this sentence from the manuscript.

Line 323: Add “e.g.,” before Childs and Parfitt

Fixed.

Line 326: Add references that show ferrihydrite is more prevalent than allophane and imogolite in Fe-rich volcanic materials

Thanks for this suggestion. This statement is related to the previous reference so we have moved the Parfitt et al (1988) reference here.

Line 328: In the discussion, further explain why the presence of ferrihydrite provides insight into early Mars history and not more modern environments. Consider defining “early history” (e.g., number of billions of years ago).

As mentioned previously, we decided to focus the discussion only on the Hesperian period and provide an approximate date of 3 billion years ago. We also note that determining the exact timing of ferrihydrite formation requires additional constraints beyond the scope of this study and we leave this for future investigations.

Line 337: Add “e.g.,” before Carr and Head

Fixed.

Line 350: The statement that ferrihydrite may have formed from the last widespread aqueous episode seems to contradict the statement that ferrihydrite formed during early Mars history.

Thank you for this comment. We have decided to remove this contradictory sentence from the manuscript.

Line 360: Add a comma after “solutions”

Fixed.

Line 366: Add “e.g.,” before Burns

Fixed.

Line 385: Change “+2” to “2+”

Fixed.

Line 398: Capitalize “Mars Sample Return”

Fixed.

Line 398: Change “bring back regolith samples” to “deliver regolith samples to Earth”

Fixed.

Line 400: Consider elaborating on what we can learn about ferrihydrite and how it formed in the returned samples. What types of measurements should be performed on the regolith? Could the short-range order of ferrihydrite tell us if ferrihydrite is pure or beginning to transform- to another, more crystalline Fe phase? Could the physical relationship between ferrihydrite and other phases tell us what has been altered to form ferrihydrite?

These are interesting questions. While many analyses can be performed as the reviewer suggests but we believe isotope studies of ferrihydrite may provide crucial clues. Fe, H and O isotopic systems in ferrihydrite can help reconstruct the aqueous geochemical conditions, water availability, and potential for habitability during its formation on Mars. For example, Fe isotopes ($\delta^{56}\text{Fe}$) record redox conditions during formation by their sensitivity to $\text{Fe}^{2+}/\text{Fe}^{3+}$ ratios. These measurements could also indicate potential microbial activity, as biological Fe oxidation produces distinctive isotopic signatures. Furthermore, O isotopes ($\delta^{18}\text{O}$) preserve information about water sources and temperature during ferrihydrite formation. This data helps determine formation conditions and provides clues about evaporation/precipitation cycles. Finally, H isotopes (δD) record information about water sources and indicate the degree of water-rock interaction, and the D/H ratios also help constrain atmospheric water loss through time. Discussion on these points is beyond the scope of this paper and therefore we only include a brief mention of the isotopes in the last sentence of the paper.

Line 403: Hyphenate “light-toned”

Fixed.

Figure 2b: Include the spectrum of basalt for comparison to the mixtures. Show spectra of mixtures with other Fe phases (e.g., akaganeite) either here or in the supplementary information.

Thank you for this suggestion. Fig. 2b now includes also a mixture of akaganeite.

Figure 5c: Label ferrihydrite peak positions in angstroms

Fixed.

. Show the search-match results in the supplementary information.

Given the distinctive and well-documented X-ray diffraction pattern of 2-line ferrihydrite, we believe the raw XRD patterns shown in the paper, with peaks labeled in angstroms, are sufficient to demonstrate phase identification. The characteristic broad peaks at $\sim 2.6 \text{ \AA}$ and $\sim 1.5 \text{ \AA}$ are diagnostic of 2-line ferrihydrite and are unambiguous in our data. In addition, we also included a reference to a review paper on ferrihydrite by Jambor & Dutrizac (1998) to support our identification.

Figure 5d: Show the LIBS spectrum for comparison to the shoulder feature in Mars data.

LIBS spectrum of dust added.

Figure 5: Could impurities in ferrihydrite contribute to the shoulder at $\sim 650 \text{ nm}$?

While ferrihydrite is known to include impurities owing to its nanoscale particle size and adsorptive properties we find it unlikely that the 650 nm feature is caused by an impurity. See our previous comment.

Figure 6: Explain the reason for the difference in the curves at high phase angles (e.g., there is a roll off in Mars data but not in the lab data)

We clarified in the caption the differences in roll off point between the Martian data and laboratory experiments.

Figure 7a: Explain the reason for the shift in the 1.4-um band

It happens when adsorbed water is present and contributes to a 1.41 micron band (which also has contributions from structural OH). The adsorbed water absorption is at longer wavelengths than the structural water, so when the adsorbed water is lost, the band minimum can shift to lower wavelengths. Similar effects have been observed in Cloutis et al. (2008) and Poitras et al. (2018). We include this statement and references in this caption.

Line 488: Hyphenate "background-removed"

Fixed.

Line 491: Hyphenate 1.9-um

Fixed.

Figure 8: Explain how the figure was made in the methods section. Also explain what "Ferrihydrite-2L" means.

We have included a short text on the pe vs ph diagram in the caption of this figure. The "Ferrihydrite-2L" was renamed to Ferrihydrite (2-line) in the diagram. In the caption we explain what is 2-line Ferrihydrite.

Line 499: Move the RMSE calculation section after the VNIR and FTIR spectroscopy section.

Fixed.

Line 500: Clarify that the spectra are VNIR reflectance spectra (e.g., “To compare the goodness of fit of our lab mixture VNIR reflectance spectra and the VNIR reflectance spectra acquired by orbiters...”)

Fixed.

Line 501: Add a comma after “rovers”

Done.

Line 518: Add a comma after “goethite”

This sentence was updated based on rev 1 suggestion.

Line 520: Clarify whether iron oxide powders were analyzed via XRD before the experiments to confirm purity and that the minerals were not altered over the course of 14 years.

Thank you for this important comment. We have conducted comprehensive XRD analyses of all iron oxide-hydroxides used in this study to verify their mineralogical purity and stability. This analysis revealed that our primary "schwertmannite" sample was misidentified and is actually akaganeite, likely due to a labeling error when the samples were originally prepared 14 years ago. We have updated Figure 2b accordingly to show the "Akaganeite-basalt mix" spectrum. Fortunately, we have another synthetic schwertmannite sample that we decided to include in this study. This second schwertmannite sample contains natrojarosite as an impurity phase, and we have included the XRD pattern and spectral data for confirmation in the supplementary file. Using this sample we also made a schwertmannite-basalt mixture in Figure 2b to address your earlier comment regarding the inclusion of additional iron hydroxide phases. The XRD patterns confirm at all other iron oxide samples maintained their original mineralogical structure and crystallinity over the storage period. Complete XRD characterization data for all phases are now provided in Supplementary Figure 15.

Line 524: Add a comma after “samples”

Fixed.

Line 524: Specify whether sieving was done dry or wet

Specified.

Line 526: Change “sample” to “samples”

Fixed.

Line 528: Fix the spelling of goethite

Fixed.

Line 531: Add a comma after “size”

This sentence was rewritten to address another reviewer’s point.

Line 531: Explain the source of the basalt in the mixtures

We explain this in the second paragraph of the methods section. The basalt sample used in this study is from the University of Bern geological collection and was sampled from an unknown locality in

Virginia, USA. However, we include here mineralogical analysis of this sample as shown in the Supplementary file.

Line 540: State the temperature under which the ethanol was evaporated

It was evaporated under ambient temperature. This information is included in the text now.

Line 545: Explicitly state the proportions of minerals and basalt in the mixtures in a table in the methods or supplementary information

All mixing ratios used in this work are now shown in Supplementary Table 1.

Line 554: I suggest rearranging the sentence to “Spectra of ferrihydrite-basalt mixtures in the VNIR and NIR range were acquired...”

Fixed.

Line 556: Specify which phases and mixtures were measured at RELAB

Fixed.

Line 559: Delete “spectral”

Fixed.

Line 560: Define the spectral resolution of the instruments.

Defined.

Line 576: Define “MCS”

Fixed.

Line 576: Define what “This” refers to (e.g., “This model...”)

Fixed.

Line 597: Hyphenate “mm-thick”

Fixed.

Line 599: State for how long the samples were exposed to martian atmospheric and radiation conditions before spectral measurements. Also state the temperature of the measurements.

This is a fair concern. We updated this paragraph to include all the required information.

Line 599: Add a comma after “atmosphere”

Fixed.

Line 601: Delete “was used”

Fixed.

Line 623: The sentence that starts on line 623 doesn't read correctly to me. I think "data." should be added after "signal-to-noise", and "reflectance" should be the start of a new sentence.

Fixed.

Line 633: Add a comma after "0 deg"

Fixed.

Line 640: Be more descriptive of "pre-" and "post-exposed". For example, "Samples that had not been exposed to Mars-like atmospheric pressure and radiation were front-loaded..."

Fixed.

Line 641: Change "sample was" to "samples were"

Fixed.

Supplementary information

Line 67: Add a comma after "mixtures"

Fixed.

Line 71: Change "comparison" to "compared"

Fixed.

Line 79: Fix the spelling of goethite

Fixed.

Line 88: Explain why magnetite is not shown in Supplementary Fig. 5.

We have included a picture of magnetite in this figure.

Supplementary Fig. 6: Explain the source of the K in the synthetic ferrihydrite and whether it affects the ferrihydrite spectral properties.

Thank you for the suggestion. Our electron microprobe analysis detected ~3 wt.% K in our synthetic ferrihydrite (Supplementary Fig. 7), which originates from the KOH used during sample preparation as described in our methods. Since K⁺ ions do not have electronic transitions or vibrational features in the visible to near-infrared wavelength range (0.35-2.5 μm), this trace amount of potassium does not influence the spectral properties of our ferrihydrite samples. The observed spectral features are dominated by Fe³⁺ electronic transitions and OH/H₂O vibrational bands characteristic of ferrihydrite.

Line 101: Add a comma after "hematite"

Fixed.

Supplementary Fig. 8: Show the martian spectrum for comparison to laboratory spectra.

We added a spectrum of Martian dust acquired by ChemCam in this figure.

Line 110: Add a comma after "basalt"

Fixed.

Line 112: Change “a goethite” to “as goethite”

Fixed.

Line 119: Describe and justify in the methods the type of Mg sulfate that was used in the mixtures (e.g., epsomite)

Fixed.

Supplementary Fig. 10: Change c and d to a and b

Fixed.

References

- Chevrier, V., Mathé, P.-E., Rochette, P., Grauby, O., Bourrié, G., & Trolard, F. (2006). Iron weathering products in a CO₂+(H₂O or H₂O₂) atmosphere: Implications for weathering processes on the surface of Mars. *Geochimica et Cosmochimica Acta*, *70*(16), 4295–4317.
<https://doi.org/https://doi.org/10.1016/j.gca.2006.06.1368>
- Jambor, J. L., & Dutrizac, J. E. (1998). Occurrence and Constitution of Natural and Synthetic Ferrihydrite, a Widespread Iron Oxyhydroxide. *Chemical Reviews*, *98*(7), 2549–2586.
<https://doi.org/10.1021/cr970105t>
- Sassi, M., & Rosso, K. M. (2022). Ab Initio Evaluation of Solid-State Transformation Pathways from Ferrihydrite to Goethite. *ACS Earth and Space Chemistry*, *6*(3), 800–809.
<https://doi.org/10.1021/acsearthspacechem.2c00026>